# Non-Random Genome Editing and Natural Cellular Engineering in Cognition-Based Evolution

**DOI:** 10.3390/cells10051125

**Published:** 2021-05-07

**Authors:** William B. Miller, Francisco J. Enguita, Ana Lúcia Leitão

**Affiliations:** 1Banner Heath Systems, Phoenix, AZ 85006, USA; 2Instituto de Medicina Molecular João Lobo Antunes, Faculdade de Medicina, Universidade de Lisboa, Av. Prof. Egas Moniz, 1649-028 Lisboa, Portugal; fenguita@medicina.ulisboa.pt; 3MEtRICs, Department of Sciences and Technology of Biomass, NOVA School of Science and Technology, FCT NOVA, Universidade NOVA de Lisboa, 2829-516 Caparica, Portugal; aldl@fct.unl.pt

**Keywords:** self-reference, natural genetic engineering, natural cellular engineering, cognition-based evolution, senome, niche construction, holobiont

## Abstract

Neo-Darwinism presumes that biological variation is a product of random genetic replication errors and natural selection. Cognition-Based Evolution (CBE) asserts a comprehensive alternative approach to phenotypic variation and the generation of biological novelty. In CBE, evolutionary variation is the product of natural cellular engineering that permits purposive genetic adjustments as cellular problem-solving. CBE upholds that the cornerstone of biology is the intelligent measuring cell. Since all biological information that is available to cells is ambiguous, multicellularity arises from the cellular requirement to maximize the validity of available environmental information. This is best accomplished through collective measurement purposed towards maintaining and optimizing individual cellular states of homeorhesis as dynamic flux that sustains cellular equipoise. The collective action of the multicellular measurement and assessment of information and its collaborative communication is natural cellular engineering. Its yield is linked cellular ecologies and mutualized niche constructions that comprise biofilms and holobionts. In this context, biological variation is the product of collective differential assessment of ambiguous environmental cues by networking intelligent cells. Such concerted action is enabled by non-random natural genomic editing in response to epigenetic impacts and environmental stresses. Random genetic activity can be either constrained or deployed as a ‘harnessing of stochasticity’. Therefore, genes are cellular tools. Selection filters cellular solutions to environmental stresses to assure continuous cellular-organismal-environmental complementarity. Since all multicellular eukaryotes are holobionts as vast assemblages of participants of each of the three cellular domains (Prokaryota, Archaea, Eukaryota) and the virome, multicellular variation is necessarily a product of co-engineering among them.

## 1. Introduction

This contribution to the Special Issue, Searching for Non-random Genome Editing Mechanisms, intends to defend the premise that non-random genome editing and other non-random genetic activities are credible and significant drivers of evolutionary development. As its means of support, a non-conventional framework for the sources of evolutionary variation is presented. It is argued that both random and non-random genetic variations are productively deployed by intelligent cells to meet environmental stresses in the defense of cellular homeorhesis and immunological self-integrity. 

Traditional Neo-Darwinian precepts have been richly modified over the last several decades but remain centered within several enduring pillars: (a) evolution is primarily due to random genetic variations; (b) such variations are subject to differential selection across a fitness landscape; (c) the resulting process of descent through modification is necessarily gradual; and (d) the target of selection is the visible macroorganic form [1,2]. The essentials of Cognition-Based Evolution (CBE) as an alternative to Neo-Darwinism have been previously enumerated [3,4,5,6,7,8,9,10,11]. CBE upholds that cellular cognition underscores biology and its evolutionary development. Life is both defined and characterized by self-referential awareness as basal cognition [3,4,12,13,14,15]. Cellular intelligence is the uniformly exhibited property of self-referential assessment of information at its scale [3,4,7,13,16,17,18,19,20,21]. Accordingly, intelligent cells can receive, assess, communicate, and deploy information to sustain individual states of self-integrity. Importantly, this process of information assessment is characterized by inherent self-reference. It is argued that it is this ‘knowing’ quality, as instantiated within the cellular form billions of years ago, that separates living entities from automata or machines. This sense of ‘self’ is crucial to biology since the defense of ‘self’ defines biological development and its immunological context. Hence, when cells sense and deploy information, it represents cellular problem-solving as its bioactive means of maintaining self-directed homeorhesis as states of dynamic flux that support cellular equipoise [3,4,7,9,10,22]. This type of discriminating self-referential assessment requires cellular measurement of both an internal status and the outward environment. Furthermore, cells abundantly communicate this measured information to other cells. This is not at all theoretical. Recent research demonstrates that during morphogenesis, cells use filopodia (actin-based finger-like protuberances from cell membranes) to coordinate angiogenesis as an active form of basal perception, time-keeping, and adaptive problem-solving [23]. It is this coordinating cell–cell communication that propels the linked tissue ecologies that enable living forms. Although there is no universally agreed-upon definition of basal cognition, in the introduction to their thematic issue, ‘Basal cognition: conceptual tools and the view from the single cell ‘, Lyon et al. [24] quote the well-recognized definition of Shettleworth: “Cognition refers to the mechanisms by which animals acquire, process, store, and act on information from the environment. These include perception, learning, memory, and decision-making” [25] (p. 5). Importantly, Lyon et al. [24] categorically assert that this definition encompasses all living organisms, including microbes and further contend that the information-processing dynamics of all forms of life are part of a ‘continuum’ with human cognition. With that background, it specifically argued that basal cognition is embodied in the cellular form. This level of cognition enables capable cellular sensorimotor functions that can be directed to cell-centered problem-solving through self-referential measurement and directed cell–cell communication. Consequently, coordinate multicellularity is the product of natural cellular engineering and cellular niche construction as cellular problem-solving. 

Four further propositions are contended. First, the phenotypic variations that are the product of differential natural cellular engineering and niche constructions are a function of largely non-random cellular problem-solving in response to environmental and epigenetic impacts. Secondly, since all multicellular eukaryotes are holobionts comprised by participants of each of the three cellular domains (Prokaryota, Archaea, Eukaryota) that interact with an interceding virome, multicellular variation is a product of co-engineering among all of them and includes viral symbioses. Third, natural cellular engineering and its coupled process, natural genetic engineering, are directed toward the continuous protection of the self-identity of each of the cellular participants that is best achieved through a collaborative form. Fourth, in contrast to gene-centric Neo-Darwinism, biology is better understood as a comprehensive and reciprocating cognition-based informational interactome that manifests as collaborative co-engineering in response to environmental stresses. This perspective enlarges on the previously introduced concept of biology as an informational interactome constituting a distributed network of reciprocating communication and interaction among cognitive organisms and their environments [26]. Importantly though, in this context of co-engineering, genes are tools of intelligent cells. 

## 2. Traditional Views on the Sources of Biological Variation 

Over the past decades, there have been many calls for significant adjustments to the Modern Synthesis and its version of adaptive variation [27,28]. Despite these critiques, the defense of that standard remains solid. A recent summary review offers a succinct encapsulation of that stance: “ the core tenet of the MS (Modern Synthesis) is that adaptive evolution is due to natural selection acting on heritable variability that originates through accidental changes in the genetic material” [29] (p. 1). Certainly, other potential causes of adaptive variation beyond random genetic mutations are acknowledged as intermittent accessories in evolution including epigenetic mechanisms, lateral transfers, transposable elements, and paramutations. However, despite this pluralism, a resolute tenet remains: “allele frequency change caused by natural selection is the only credible process underlying the evolution of adaptive organismal traits” [29] (p. 10). In this framework, these allele frequencies achieve biological expression through ‘adaptive’ peaks’ and ‘phenotypic valleys’ through processes that are intrinsic to the genome and random with respect to fitness. This concentration on the central primacy of random genomic mutations has led to the general dismissal of the importance of lateral transfers in eukaryotes as too rare to be consequential [30]. Such orthodoxy is not exceptional. Charlesworth et al. emphasize: “… neo-Darwinian evolution requires the transformation of a population over time as a result of natural selection. If variants tended systematically to arise when they are adaptive, many or all individuals in a population could acquire adaptations without the need for selection; this would indeed constitute a serious challenge to the MS. As John Maynard Smith once said ‘… the question of the origin of hereditary variation remains central to evolutionary biology, if only because Lamarck’s theory is the only alternative to Darwinism that has been suggested’” [29] (p. 2). 

Even so, some substantial modifications of Neo-Darwinian variation theory have been previously argued. In the 1930s, Otto Schindewolf and Richard Goldschmidt individually championed the saltational theory of evolution which asserted large leaps between species due to macromutations that create coordinated adaptive phenotypes [31]. Gould and Eldridge articulated their theory of punctuated equilibrium with rare bursts of evolutionary change that rapidly separate species interrupting long periods of species stasis [32]. Many have urged an Extended Evolutionary Synthesis as a modification of the Modern Synthesis by emphasizing additional sources of variation including epigenetic inheritance, niche construction, and biased development [33]. Moczek recently argued that evolutionary developmental biology (evo-devo) offers to remake evolution by concentrating on developmental bias, facilitated variation and the ability of living systems to direct their own formation beyond selection. Within that construct, two salient points are asserted: “The metaphor of genes and genomes as blueprints or programs for organisms and their parts have outlived their usefulness long ago…” [34] (p. 24) and, “Environmental conditions provide, one way or another in all organisms, information critical for the completion of normative development” [34] (p. 25). 

Cognition-Based Evolution (CBE) accommodates those perspectives by offering an entirely new framework in comparison to conventional Neo-Darwinism. It directly asserts that evolutionary biology is self-referential cellular dynamics as an informational interactome. The basic living unit is an intelligent measuring cell that exchanges information with other cells through an obligatory cellular information cycle based within thermodynamics. Beyond random genetic mutations, a wide variety of exogenous and endogenous genetic impacts and cytoplasmic modifications coordinate to substantially influence evolutionary outcomes [7,9,10]. Trerotola et al. provide an extensive review of genome-wide association studies of physiological traits, phenotypic variations, and a variety of genetically associated diseases [35]. They found that intrinsic central genetic mutational errors or allelic polymorphisms can account for only a small proportion of observed heritable variation. Instead of Neo-Darwinian chance as the source of biological variation, CBE places variation as the product of cellular measurement which permits co-engineered outcomes from participants across the cellular domains as collaborative ecological units [6]. From a large variety of informational inputs, self-referential cells measure and collaborate in multicellular ecologies to consistently meet environmental challenges [7,9]. 

## 3. Unicellular Insights into Biological Variation 

Extensive basic research has been conducted on prokaryotes that help to elucidate the general evolutionary pathways that account for variation across the living spectrum. Experiments with prokaryotes under differing conditions of stress illustrate each of the fundamental concepts of biological variation that equally pertain to the eukaryotic realm [36]. One example is the transfer of antibiotic resistance genes between donor cells equipped with a bacterial resistance gene and recipient cells. Within biofilms, participants transfer information among themselves along multiple paths [37,38]. Within that collective form of life, each microbial participant self-assesses its individual status as part of a collaborative appraisal of environmental stress [39]. This co-linked set of actions is sustained by quorum sensing as a ubiquitous feature within biofilms. Quorum sensing is a complex system of cell–cell communication primarily via chemical signaling molecules that enables collaborative behaviors among cells and the trading of resources [40]. In the case of antibiotic resistance, genetic material is transferred among participants as information that can be deployed to meet an antibiotic threat. In the collaborative biofilm structure, participants with antibiotic-resistance genes will indicate their availability to other cells [41]. The donor cell and the recipient cell attach via a pilus that enables the donor-to-recipient transfer (Figure 1).

The complexity of this interaction and the fact that it is individualized and negotiated reveals that it is far more than a simple exchange among automatons. Each participant in this exchange must be aware of its own circumstances within the context of the further multicellular collaborative. This requires a complex sense of self-identity that includes ‘self’ and a communicating ‘other’ which extends beyond an immunologically foreign organism. Therefore, such a highly complex, consensual, and reproducible interchange requires an entire suite of elemental capacities. Most particularly, these exemplify the basal cellular characteristics of cooperation and interdependence that distinctly extend beyond any alternative narrative of simple competitive drive [7]. Clearly then, a prokaryote can engage in the individual self-referential assessment of information and deploy its tools to meet environmental stresses. Further yet, as this exchange embodies, genes are tools that can be traded and exchanged [9]. It defaults that these reciprocating activities rely on measurement and prediction. The functional meaning of this ensemble activity can be productively reduced to the cooperative engagement between self-aware participants that directs towards mutualized problem-solving. As such, it represents an explicit example of natural genetic engineering. Further, it is argued that this demonstrable non-random natural genetic engineering is a sub-type of a larger construct of natural cellular engineering that enables ubiquitous biofilms and reiterates as holobionts. 

Plasmid transfer from prokaryotes can also be effected across other domains, and further, need not be entirely mutualizing. *Agrobacterium tumefaciens* is often described as a natural genetic engineer and will transfer genetic material to plant cells [43]. A Ti plasmid contains the relevant *vir* genes that are required to mediate the process of DNA transfer and enable the expression of transferred T-DNA in plant eukaryotic cells. One transferred set of genes encodes proteins that affect plant cell growth regulators and a second set stimulates the synthesis of small molecules that are used by *A. tumefaciens* as a source of carbon and nitrogen. The exact transfer mechanism is incompletely understood and it is not enacted via a pilus. It has been established that it requires an intricate process that involves the synthesis of complex membrane fibrils from prokaryotes that permit plant cellular attachment [43]. No matter the exact mechanism, this represents another form of problem-solving at the unicellular level. *Agrobacterium* cells use their sensory apparatus to detect plant-derived signals and respond by modifying the transcription level of their own *vir* genes in complex synchrony. When the infectious transfer is complete, the virulence system must be silenced by down-regulation of that same elaborate *vir* gene-based system. Further, the integration of the transferred DNA which enters the plant genome yields non-random biological results that directly serve the bacterium doing the transfer. Matveeva and Otten have found that HGT from *Agrobacterium* to monocots and dicots is a widespread form of natural genetic engineering with transferred genetic potential expressed in some plants and silenced in others [44]. 

At the level of the cell, responses to environmental stress can involve a form of internal, intra-cellular engineering. Immotile strains of *Pseudomonas fluorescens* have been human-engineered in the lab to lack flagella due to *fleQ* deletion, a critical regulatory gene [45]. These flagellar-deficient bacteria were able to regain their flagella and use it for motility within 96 h. They can re-purpose alternative genetic pathways to increase the expression of a nitrogen regulatory protein, NtrC, which stimulates alternative target genes, boosting their expression and returning bacterial motility. This is cellular problem-solving with complex internal re-wiring that represents a form of internal engineering. Marshaling such coordinated genetic variations to regain an intricate capacity within the space of 96 h cannot be the result of a random process. Instead, it is argued that these cells are self-aware and can direct available tools to problem-solving, including genes. Moreover, just as with human engineering, the solution comes with a cost. Improved motility directly correlates with a deterioration in nitrogen regulation [43]. In this process, an environmental trigger is confronted and internalized to continue essential cellular functions at some cost. A similar pattern of reactive response has been previously described as accelerated target mutagenesis in *Escherichia coli* as an adaptive stress response in which mutations cluster in targeted genomic foci as a form of mutagenic repair [46,47]. 

It is not conjectural that cells are capable of self-directed engineering. Indeed, that capacity is being put to use by human bioengineers to refine their search for beneficial natural products. A research group stumbled on a routine process that bacteria use for adaptation [48]. The project was directed towards finding new versions of rapamycin, a naturally occurring compound produced by bacteria and some fungi to defend against pathogens. It is also used medically to treat some forms of cancer and prevent organ transplant rejection. A temperature-sensitive replicon (a specific region of DNA or RNA that replicates from a single genetic origin) was inserted into *Streptomyces rapamycinicus* in an attempt to rationally bioengineer the rapamycin biosynthetic gene cluster (BGC). It was expected that the introduction of a single focused replicon would yield only a single new strain of *S. rapamycinicus.* Instead, a wide range of unexpected analogs (rapalogs) was produced in surprising amounts. Each was produced by some, but not all cells. Each of these compounds demonstrated genetic variations confined to the polyketide synthase region of the rapamycin BGC and each affected a different structure of the polyketide chain backbone. Variants in the rapamycin gene cluster have been traditionally ascribed to genetic duplications, random mutations, homologous recombination events, and horizontal gene transfers that are thought to have occurred over a traditional evolutionary timescale. In contrast, it is believed that this replicon initiated a critical genetic instability that yields a form of accelerated evolutionary bioengineering. The interaction inadvertently revealed that a single genomic event can lead to a plethora of adaptive variants from localized recombinations and deletions exclusive to a single genetic region. This pattern of concentrated recombination is now recognized as a form of natural genetic engineering used by cells to diversify polyketide structures [49]. Consequently, it can be argued that this rapid rapalog diversification can be regarded as a constrained exploration of potential genetic sequence variants to confront an environmental introgression. 

Collective biofilms represent a predominant form of prokaryotic life since they permit collective engineering as an effective form of cellular problem-solving. In *Bacillus subtilis* biofilms, the participating cells can sub-specialize to elaborate an optimal extracellular matrix for either cell surface adhesion or colony mobility [50]. The specific composition of this extracellular matrix must differ to meet exact environmental and surface conditions to permit either adhesion or motility. To produce both, constituent cells partner in specialized activities that completely differ from those that they exhibit within their free-living form [51]. These are consensual actions. There are high levels of traded resources and some participants accept the voluntary loss of certain cellular functions to support the collective whole [52,53]. Each of these united cellular forms with their exclusive and different outputs represents a type of phenotype and each variety has its differential discrete function, whether to stick or to migrate. 

To effect any of these cellular outputs requires a complex process of internal self-directed assessment of environmental cues. It has been previously asserted that this reception and assessment of environmental information proceeds through a cellular senome [54]. The senome is conceived as the total summation of all of the sensory inputs of any cell which can be derived by the entire sensory apparatus and sensory tools of the cell [9,54]. Cells maintain their homeorhesis through the totality of their senomic inputs. Prokaryotes have highly developed sensorimotor circuits that link throughout the cell connecting acquired information to adaptive behaviors [20] that proceeds through a definable cellular information management system [3,4,6,7]. Therefore, the cellular senome is both a cognitive gateway and a biological nexus through which sensory memory is encoded into bioactive molecules which is fully active across the cellular domains. It is argued that this concatenated process of senomic integration should be regarded as intracellular engineering. This senomic informational matrix, as the ground state of individual unicellular self-identity, aggregates in multicellularity as a common informational platform that energizes multicellularity either as biofilms or holobionts. It follows that this intracellular reaction set, with all of its convoluted ramifications, should be viewed as the actual source of biological variation since, perforce, all phenotypic variations must initiate at the cellular level. 

## 4. Variations in Multicellular Organisms

### 4.1. General Mechanisms 

Villarreal and Ryan [55] declare that evolutionary change cannot begin with natural selection [55]. Darwin would have insisted similarly since any variations upon which selection might act must come first. Villarreal and Ryan [55] indicate that there are four major mechanisms for changing heredity: mutational copying errors, epigenetics, symbiosis, and hybridization. Notably, they emphasize that viruses may be involved in several of these mechanisms, most particularly through horizontal genetic transfer (HGT). In eukaryotes, viral transfers and transposons are critical to gene regulatory networks as a source of possible new genetic variations that are considered critical to the origin of the immune system, perhaps based on superinfection exclusion mechanisms [56]. Significant evolutionary transitions in eukaryotes have been linked to the non-random distribution of transposable elements that represent co-adaptive strategies between transposons and cells [57]. Further yet, based on common properties and phylogenetic relationships, it has been proposed that transposons may be the origin of those regulatory genetic networks that have contributed many of the fundamental properties of eukaryotes [58]. 

Many other potential mechanisms for genetic variation beyond replication errors have also been uncovered. Caporale and Doyle present evidence of ‘biased’ mutations, stressing that these are non-random mutations that target certain genetic clusters that are mutational ‘hot spots’ [59]. Others have argued for facilitated variation in which genetic changes that lead to phenotypic variation are concentrated in a subset of DNA that constitutes its regulatory elements [60]. In this framework, new traits arise from recombinations within more limited genetic subsets. In effect, ‘facilitated’ variation proposes that Neo-Darwinian mechanisms are valid if the genomic field of view is sufficiently constrained through a reduction of the applicable genomic subset. That argument proposes that if the size of the genome in which random events might supervene is sufficiently constricted to ‘core processes’ that function in defining development and physiological functions, then random genetic mutations within this narrowed genomic sphere could more readily lead to consequential heritable regulatory changes. Although the theory intends to defend Neo-Darwinian tenets, casual consideration indicates otherwise. Simply put, the ‘core’ becomes a set of regulatory modules that is separate and distinct from others. In that case, variation is biased and no longer random as it affects some genomic foci and not others to unequal degrees. Indeed, this contradiction has been noted and even embraced as ‘developmental bias’ as a form of directed non-random phenotypic variation among proponents of the Extended Evolutionary Synthesis [33]. 

Some urge a consideration of an expanded view of the genome beyond its direct coding function. The genome has been rethought as a flexible palette through which organisms can ‘co-construct and co-evolve with their environments’ [28]. A few have gone further, urging that the entire conception of the genome as only read-only programmatic code must be reconsidered. Shapiro champions a flexible ‘Read-Write’ genome in which cells react to environmental cues by forming nucleoprotein complexes and reconstruct function due to epigenetic formatting or by changing DNA sequence structure [61]. These non-random actions are written into cellular genomes as genomic inscriptions through symbiogenesis, horizontal genetic transfer, and natural genetic engineering. In consequence, mobile genetic elements disperse through the genome to provide coordinating network signals. This acts as a non-random mechanism for the direct internalization of cellular stresses. Phenotypic adaptative expression can result, scaling primarily with non-coding DNA and with non-coding RNAs as abundant repetitive elements [62,63]. Employing the perspective of biocommunication, Witzany has also integrated natural editing of the genetic code with epigenetics in context-dependent patterns, further maintaining that viruses also adjust their symbiotic relationship with cells through natural genetic editing [63]. 

DeLoof has insisted that evolution should not be considered a ‘noun’ but instead should be modeled as a ‘verb’ [22,64]. Through the contemporary perspective of natural genetic editing as a ‘Read-Write’ genome, evolving through genomic reciprocation with the environment and epigenetic pathways, the concept of the genome has been substantially revised from a passive repository of code to an active and pliant participant in evolutionary development. 

### 4.2. The Tools of Viral-Cellular Variation 

A multitude of sources of genetic variation have been identified that influence the lives of cells and contribute to their relationships with others. Their magnitude and intricate complexity mandate that there are non-random cell-wide coordinating pathways that permit productive biological outcomes. 

Some intercellular processes with significant genetic consequences are cell-wide rather than representing localized or compartmentalized genetic phenomena. In the process of endocytosis, one cell can engulf another and acquire a large number of bioactive molecules or genetic material. The targeted material is transported to the cell interior via a cellular membrane-lined vacuole that contains both information and nutrients. For example, breast cancer cells can acquire lipoproteins which alter gene expression through receptor-mediated endocytosis [65]. Viruses make much of this process through a variety of well-elaborated mechanisms for intracellular entry that includes endocytosis among many others [66]. Additionally, it is well-recognized that endosymbiosis accounts for the origin of Eukaryota [67,68]. It is believed that eukaryotic mitochondria and plastids are the residua of engulfed bacteria that became indwelling endosymbionts as useful organelles. Through a process of genome reduction, some of the engulfed genetic material enters the host central genome via horizontal acquisition [69]. 

Entosis is another large-scale source of variation among cells. Complete engulfment or invasion of one living cell by another as entosis results in the active transfer of the entire informational-genetic architecture of one cell by another [70,71]. Surprisingly, among non-neural organisms, this accompanying transfer of an entire suite of available resources is so complete that it can even include learned patterns of behavior [72].

Once a disputed source of genetic variation and novelty, horizontal gene transfer (HGT) is now accepted as a common method of information transfer among unicellular organisms. HGT has played a major role in unicellular genomes and it is estimated that as many as three-quarters of bacterial genes have undergone at least one HGT event at some point in evolution [73]. Although previously thought to be rare among eukaryotes, further research has confirmed that it is not uncommon, especially from transposable elements. Cell to viral transfers are common, viral endogenization is very frequent in some eukaryotes, and viruses serve as vectors of genetic materials between eukaryotes [74]. It is now well-established that these transfers have adaptive value and promote their own dissemination within the gene pool of an active mobilome. Significantly, it has been found that the evolution of large multicellular eukaryotes is actively influenced through HGT from microbial associations that may participate in metabolic innovations [75]. 

The impact of endogenized retroviruses on evolutionary development is firmly established. However, it is not only retroviruses that can be endogenized. All types of eukaryotic viruses can be integrated into a recipient heritable genome [76]. The domestication of their endogenized sequences has stimulated new genes [77]. Some viruses are resisted, but others are recruited for physiological development, such as for the formation of the numerous syncytiotrophoblastic tissues and the placenta [78]. Thus, viral insertions are dominant drivers of evolutionary adaptive change, are a significant percentage of the human proteome conserved in mammals, and contribute to genome plasticity [79,80,81,82]. 

It is believed that many of these transfers target regulatory genomic components. Such retrotransposition events can participate in new regulatory regions and shape patterns of gene activation in the cell as sources of evolutionary innovation [83]. Necessarily then, they interact with any pre-existing genomic context within cells. Some believe that these retrotransposition events are not random but are, instead, targeted interactions with ribosomal RNA genes activated during infectious interchanges [84]. These infectious epigenetic impacts are highly consequential and continuous throughout evolution with significant retrotransposon alteration of most eukaryotic genomes affecting gene expression in animals and plants [85].

Endogenous retroviruses (ERVs) are present in the genome of all vertebrates, originating from germ-line infections [81]. ERVs are recognized contributors to genome plasticity such as in the development of the mammalian placenta [86]. Horizontal transfers exist not only in prokaryotes, but between prokaryotes and multicellular eukaryotes. Recent research on the human microbiome confirms the importance of the microbiome for metabolism and immune function, but also indicates the possibility of direct genetic transfers [75]. That research suggests that recent genetic transfers from the human microbiome by HGT is 25-fold more frequent from human-associated microbial populations than others, and such exchanges are estimated to be 50-fold greater between eukaryotic cells within specific tissue ecologies and their directly associated microbiomes [75]. The range of these epigenetic impacts is suspected to be substantial and enduring. It has now been shown experimentally that there is not only frequent heritable transmission of bacteria between eukaryotic generations, but that non-chromosomal epigenetic hereditary variation secondary to that microbial cohort, such as immunoglobulin-A levels, is also transmissible and can mimic the effects of chromosomal mutations [87].

Estimates suggest that up to 30% of all human protein adaptations that account for the divergence of humans from chimpanzees have been derived from viruses [82]. This has a significant further implication since such viral impacts need not be simply random. Research indicates that the genetic effects due to viruses do not conform to a simple stochastic distribution [88]. Even further, at least some HGT events are not random. Goldenfeld and Woese detail such an instance with the conjugative plasmid transfer of antibiotic resistance via the plasmid pCF10 in *Enterococcus faecalis* [41]. This process is controlled by coordinate cell–cell communication that arises from recipient cells lacking antibiotic-resistant plasmid-based genes that carries to plasmid-carrying donor cells.

The acquisition and modification of genomes by means of non-vertical transmission, once thought to be rare, is now accepted as a crucial factor in evolutionary development both within the prokaryotic and eukaryotic realms as a highly integrated mobilome of genetic exchange among the domains [89]. The contemporary view of the genome is as an ecosystem in which diverse communities of transposable elements (TEs) take residence in patterns of non-random distribution [90]. Further, these non-random actions proceed along familiar ecological terms including both cooperation and competition [91]. It is now believed that TEs occupy a larger fraction of the genomic DNA in eukaryotes than prokaryotes, estimated at 70% of the genome in vertebrates and over 80% in plants, representing a source of horizontal expansion among genomes [73]. Pertinently, there is a sizable opinion that these transfers and their insertion sites are not random occurrences [57].

In this way, retrotransposon transfer and activation is frequent and is a contributing factor to the functional regulatory machinery of the cell in the face of environmental stresses [12]. Given their transfer between cells, retroelements can be considered residues of previous infection interchanges from which phenotypic variation can eventually emerge [92]. Consequently, HGT as an active mobilome can be regarded as a major source of genetic variability and novelty and a major force propelling genomic variation and biological innovation in eukaryotic genomes in non-random patterns [93,94,95,96]. Their role as major players in the evolutionary development of plants, especially their effect on the induction of polyploidization has been emphasized [97] as well as their participation in gene amplification in metazoans [98]. For example, evaluation of the genomic location of TEs in fungal genomes found that younger TEs that are more likely to be active will non-randomly cluster with one another away from other genic regions, possibly indicating a site preference [99]. Research in *Escherichia coli* indicates that transposon-mediated mutations are more frequent when beneficial than detrimental [95]. Thus, the non-random distribution of TEs in prokaryotic and eukaryotic genomes is the consequence of both TE integration site preferences and selection for phenotypic solutions to environmental stresses [100]. 

It has been proposed that the impact of TEs might be amplified by massive bursts of activity that result from environmental impacts [101,102]. These represent episodes of stress-induced radical genomic rebuilding through sequential domestications, polyploidy, interspecific, or intergeneric hybridizations. The TE-Thrust hypothesis, supported by an analysis of primate genomes, asserts that TE-facilitated processes engage in genomic self-engineering, shifting genomic regulatory elements to permit beneficial adaptive responses [103]. These same bursts of TE activity have been proposed as a rapid mechanism of adaptation promoting structural variations to explain the swift adaptation of invasive species to novel environments [104]. As an example, there are an estimated 500,000 repeats in LINE1 in the human genome. Although once considered merely junk DNA, or potential genomic parasites that randomly gain access to a central genome, recent research contradicts this prior viewpoint. LINE1 is crucial for embryonic development after the two-cell stage [105]. It has also been found to be crucial in human brain cell function, participating in human health and disease [106].

The relationship of TEs to biological outputs and variations involves complex interactions with non-coding RNAs that are also subject to transfers. Such non-coding RNAs originate from mobile elements. Through this mechanism, new genes express across transposon sequences and participate in network regulation. Non-coding RNAs are transcribed into molecules as part of protein complexes whose non-random processing has reciprocal regulatory effects on transposons [107].

Consequently, TEs influence the genome and its transcription. They also contribute to epigenetic variations that have phenotypic consequences. That involvement expresses through molecular mechanisms and proteomics, promoting genetic exaptation, genomic regulation, and formation of retrogenes. All of these exert a collective impact on genomic plasticity [96]. Importantly, through their functional impact on adaptive variation, TEs assist organisms to meet environmental challenges by helping to shape immunological status including resistance to pathogens. Waves of expansion of TE families have contributed to the repertoire of transcription factors and cis-regulatory sequences in mammalian genomes [108]. Experiments in the model organism, *Drosophila*, confirm that TEs can modify the transcriptional pattern of host genes to contribute to short-term adaptations of physiological capacity to environmental stresses [109]. It is now believed that centromeric chromatin in most eukaryotes is composed of highly repetitive centromeric retrotransposons that participate in cell division [110]. Centromeric DNAs are crucial to chromosomal segregation and cell division and are not highly conserved phylogenetically. Instead, they are widely divergent among species, and even vary greatly among closely related species. 

Similarly, the wide variety of small RNAs, plasmids, and other non-chromosomal cellular genetic constituents constitute a substantial fraction of the entire genomic complement of cells. There is no doubt that RNA-mediated processes including small RNAs, RNA stem loops, and microRNAs (miRNAs) are crucial participants. Cellular life is fundamentally dependent on rRNA, tRNA, and mRNA for DNA replication [111]. Messenger RNAs (mRNAs) are potent epigenetic sources. Other microRNAs and non-coding RNAs (circRNAs, endo-siRNAs, piRNAs, antisense, and long ncRNAs) are integral aspects of cellular adaptation as sources of evolutionary novelty participating within a complex regulatory apparatus [112]. Non-coding RNAs (ncRNAs) that include a variety of small interfering RNAs (siRNAs), and piwi-interacting RNAs (piRNAs) contribute to genetic silencing and virus interfering RNAs (viRNAs) have a role in immune responses in the protection against pathogens [113,114]. All of these represent a “vast trove” of vital information for cells that is not initially derived from vertical transmission and represents a driving force in evolution [79,114]. Importantly, infective RNA networks all have a crucial role in natural genetic engineering. These help to create novelty and effect evolutionary transitions through intrinsic cooperation, community associations, and de novo nucleotide sequence insertions or deletions [12,63,114,115,116]. RNA networks of all types, such as group 1 or group 2 introns, retrotransposons, retroviruses, RNA viruses, LINEs, and SINEs, can enter and integrate into chromosomal DNA. RNA-protein networks are derived that bypass translation to participate in RNA regulatory functions [114]. Thus, all cells depend on extensive and highly coordinated RNA-mediated processes for transcription, translation, and genomic regulation as an integral part of the information storage and communication system of cells [92]. 

A variety of extra-chromosomal circular DNAs, microRNAs, and circular RNAs are also consequential cellular genetic constituents. Widespread across species, these produce short regulatory RNAs that function in non-random gene expression [117]. For example, there are abundant extrachromosomal circular DNA strands in yeast and animals and some circular DNAs function as retrotransposons and contribute to epigenetic effects, particularly in plants [118]. Diverse circular RNAs are also present in cells and participate in genomic expression and as transcription regulators [119]. Functioning at multiple levels of gene regulation, these participate in epigenetic silencing and mRNA stability [120]. Through a complex network of complementary endogenous RNAs (ceRNAs), circRNAs compete with miRNAs to influence target RNAs in both health and disease [121].

The proteome also contributes to genetic regulation. Krüppel-associated box domain zinc finger proteins (KRAB-ZFPs) are the largest family of transcription regulators in higher vertebrates. These link to varieties of TEs as participants in their silencing and derepression and as partners in species-specific regulatory networks [122]. This process, which is vertebrate-specific, plays a role in cell proliferation, differentiation, apoptosis, and cancer [123]. 

Along with KRAB-ZFPs, there are more than 1300 virus-interacting proteins that have been identified that account for a significant proportion of protein adaptations in humans and other mammals [82]. Through variable fold-conformations, virus-interacting proteins can yield multiple phenotypes that do not directly relate to cellular genetic complement. For example, Bacteriophage λ is a virus that infects bacteria by exploiting various membrane proteins through well-characterized pathways [124]. Research indicates that evolutionary differences in variable folding conformations of isogenic proteins have contributed to λ’s ability to exploit an additional host receptor for infection despite there being no difference in their genetic sequences. Because the protein can take on two shapes, this genotype can have two phenotypes, either of which can be filtered by selection. 

Another source of cellular adjustment to environmental stresses is structural variations of DNA and RNA architecture. Chromatin is an important regulator of gene expression [125] that links to the molecular determinants of cellular responses to stress and complex phenotypes in all organisms including humans [126]. These variations can result in deletions, duplications, copy number variants, inversions, or insertions that can occur in either coding or noncoding regions of the genome and can even involve transposons [127]. 

Stem loops can be an additional source of structural genetic variation. These have been identified in both DNA and RNA, occurring when two lengths of nuclei acid sequences are juxtaposed in a hairpin loop configuration with complementary nucleotide sequences that read in opposing directions. These have a copious functionality in RNA communication, serving as catalytic drivers of intracellular processes as signals and serve as building blocks of ribozymes and messenger-RNAs [111]. 

Prions are self-replicating transmissible proteins that can change conformation, act in concert, and are a well-known cause of neurodegenerative disease. They are also capable of participating in adaptive memory and as a source of heritable information [128]. In that latter regard, prions have been demonstrated to confer evolvability and phenotypic plasticity in yeast [129]. 

It is beyond the scope of this paper to consider the entire influence of the microbiome on natural genomic engineering. It is no longer disputed that our partnering microbiome contributes essential aspects of metabolism and adaptive phenotype whose range of influences extend from conception onward [130]. Given that there are estimated to be 100 trillion microbes associated with us as holobionts, their cumulative genetic complement massively outweighs our own. Their effects on the human gut [131], brain and central nervous system [132,133], and immune system are documented [134]. It is further acknowledged that the cumulative role of the microbiome is of sufficient magnitude that it should be considered a developmental organ in its own right, participating in developmental plasticity, genetic and extragenetic inheritance, and niche construction [135]. Further, there is accumulating bioinformatic evidence that the virome is a vital intercessory in abundant dialogue among the cellular domains. Viruses swap genes among a great number of organisms beyond their common hosts. These infections contribute to the diversity of genetic tools in cells that they are not typically known to infect as virus hallmark genes [136]. 

The effect of environmental stresses is also now understood to have a substantial impact on heritable genetic expression. For example, there are documented non-random transgenerational effects from starvation [137], hypoxia [138], high fat or low protein diets, toxins [139], and generalized stress [140].

The cell is an information management system and each source of variation has its individual effects. The genetic complement of any organism or any cell is influenced by a large, diverse, efficacious, and reciprocating array of inputs and adjustments. Their coordination yields productive and environmentally responsive outputs that should be regarded as mechanisms of active natural genetic engineering. Within that process, non-random genome editing contributes to adaptive solutions to cellular problems. It follows from this context that the central genome is not a sacrosanct compartment as had been initially supposed. Instead, it is a reciprocating participant in cell-wide activities that are devoted to a continuum of cellular problem-solving. In such a framework, genes of all provenances are directly acknowledged as tools of cellular faculties [7]. Every aspect of cellular life including its entire genetic complement participates in information gathering, storage, and deployment. The extracellular matrix contributes its own active role. It is a crucial player in intracellular information transfer and the maintenance of cellular integrity [141]. This matrix reciprocates with the entire genetic complement of the cell in a dynamic system of interacting protein domains. Their further evolution is dependent on other intracellular processes that integrate mobile genetic elements, contribute to exon shuffling, and by cross-species communication through prolific non-coding RNAs [9,12,62,63,115,142]. Each of these processes represents instances of genomic rewriting directed towards cellular adaptation that contributes to the regulation of eukaryotic multicellular development and reciprocates at all levels. Transposons are known to participate in this process and are, in turn, tightly regulated themselves. This new information necessitates a shift in perspective. In the traditional view, mobile genetic elements have been regarded as mere epigenetic parasites that must be suppressed through down-regulation, In contrast, TEs and their derivative transponsases are now regarded as essential elements of the evolvability of eukaryotes that contribute to the creation of genes and their ultimate expression [143]. In reciprocation, many have asserted that genomes can participate and shape their own evolution in response to environmental stresses in a manner that extends beyond random occurrences [7,9,62,116,142,144,145]. In this manner, our contemporary appraisal of the genome has itself evolved from a genetic sanctuary to our current understanding of a genome as a transitory ‘snapshot’ of a dynamic genetic mobilome [146].

## 5. The Origins and Extent of Natural Cellular Engineering

How selfish soever man may be supposed, there are evidently some principles in his nature which interest him in the fortune of others. Adam Smith.

There is no doubt that cells can measure. However, defending the proposition that collaborative cells can engineer non-random outputs through measurement requires an explanation of why they do so. Cells must measure information since all biological information is ambiguous. For self-referential cells, just as is true at our scale, the available information that is used to maintain life-sustaining equipoise is imprecise [3,4,7,147]. Simply put, if the information that was available to any cell was perfect, there would be no need for its measured assessment [9]. 

The reasons why biological information is ambiguous have been explained in detail in prior publications [4,7,9,147]. In brief summary, multiple specific restrictions compromise unfettered information quality in the living frame. The first is thermodynamic. Both the self-referential assessment of information and its communication require work. However, the conversion of energy (information) to work (information assessment and its communication) that can never be 100% efficient in any frame is amplified in any living system. Since any information assessment requires work as energy conversion, and since matter and energy are interconvertibles in physical systems, it has been argued that information is always physical [9,148]. Although debatable, in the context of biological systems, information achieves physicality as a transfer of either energy or material that must become bioactive molecules. This directly leads to a derivative limitation of information quality. Living systems require the boundary architecture of membranes. It is only under boundary conditions that the First Principles of Physiology (negentropy, chemiosmosis, homeostasis) can be sustained [149]. The transmission of energy or molecules as information/communication must go through multiple types of media and membranes which yields an obligatory degradation of information quality [147]. Each of these informational cues is also subject to variable time delays. Further yet, communication is often a broadcast among cells. Typically, neither the sender nor the receiver is necessarily known among specific collections of cells, representing another source of uncertainty, particularly with respect to pathogens. 

An additional obligatory source of informational uncertainty for cells is that the living frame is self-referential. Under those particulars, it defaults that all the information that any cell possesses is actually self-produced. Although energy and matter exist external to cells, the cell can only know anything about these informational sources through a process of self-generated appraisal. This issue has well-described as the concept of info-autopoiesis which asserts axiomatically that any information that a cell possesses is self-produced [150]. Indeed, this is precisely why the cell is a measuring instrument. It utilizes its senome (its entire sensory apparatus) in its own appraisal of any material or energetic input [55]. By definition then, for living entities, there can never be a pure data information point. This requisite directly leads further sources of information insecurity. In the observer/participant plane of reference, information is ‘round’ in the sense that for any source of information, there is always more information available for assessment than can be detected by any individual observer/participant. Information has binary characteristics that are constituted by attributes that are both observable and ‘hidden’ to any specific self-referential observer. Some information may be obscured to one observer yet available for assessment by others [4]. For example, in the case of a ball, two different observers looking at the same ball would see necessarily see slightly differing aspects of its surface and slightly different portions of the rest of the ball would be hidden from appraisal, a concept that has been termed antipodal information [7]. Hence, each observer derives slightly different meaning from the same observation. As self-referential observers, this phenomenon pertains to cells. Any cell will appraise information based on its individual position and context. To do so, cells measure self-produced information based on an entire discriminating sensory apparatus (senome) that is complex and widely distributed throughout any cell. This yields a further source of informational ambiguity. Since the measuring apparatus in cells is dynamic and not fixed, measurements will vary based on intracellular dynamics, i.e., a living cell is varying as it proceeds from measurement to measurement, even during the same stimulus as a source of inevitable noise. 

Thus, the permanent struggle of life can be defined as centered within a need to optimize information. Living organisms are required to live through an attachment to biological information but must cope with its explicit imprecision. There is no perfect information and there is always an element of doubt as interpreted at any scope and scale [7]. It is asserted that life and its myriad variations are governed by this elemental principle. 

From this background, the predominance of multicellularity within our biological system clarifies. In order to sustain their individualized states of homeorhetic equipoise, cells communicate and cooperate to improve the validity of ambiguous information. Cells assess information both individually and as a collective ‘wisdom of crowds’ [9]. This is the bioactive expression of the cellular drive to optimize its effective information. The impulse to do this begins within the self-referential frame. There is a self-organizing and self-reinforcing information cycle among all participant/observers [3,4,7]. When any living entity receives information and assesses it as a measuring function, it necessarily expends energy as a thermodynamic requirement. Necessarily, this energy becomes an available energetic signature as a source of information and communication to some other observer/participant within the same information space [4]. Since this living interplay is based on information, measurement, and communication, this obligatory shared informational interactome propels collective measurement. It is this process, with its obvious derivative benefits of energy optimization and maximization of cellular resources, that drives natural cellular engineering (Figure 2). 

Environmental cues from information space impact the senomes of individual cells, initiating a reiterating senomic information cycle. Since all cellular information is ambiguous, the self-referential assessment of information requires its measuring assessment. This process is work that becomes a signaling communication to any other self-referential organism irrespective of whether or not it is purposeful. The further cellular reception of that obligatory communication necessitates its own measuring assessment as work. This becomes a signaling further communication that reiterates as a self-reinforcing work channel. It is this process that energizes communities of cells to assess information in the aggregate which enacts multicellular phenotypes. 

In the living state, the transfer of energy is information whose reception and assessment oblige work which is automatically further communication to other observer-participants. When that communication is received as information, the cycle reiterates. The basis of the living state rests on this obligatory information cycle that expresses as natural cellular engineering. Through this pathway, cells are able to engage in productive natural cellular engineering and mutualized cellular niche constructions which form the multi-species cellular tissue ecologies that link to become holobionts. It defaults that it is from the coordination and linkages of these types of tissue ecologies that phenotype emerges [3,6,151]. When biological and evolutionary development is centered within this obligatory self-reinforcing frame of natural cellular engineering, it clarifies that the source of variation can be identified as deriving as the product of collective differential assessment of ambiguous biological information by networking intelligent cells. Several essential characteristics of cellular life define the parameters of natural cellular engineering. An ensemble of critical attributes needs to be deployed in concert. These are the cellular faculties of collaboration, coordinated cooperation, co-dependence, and mutualizing competition. These are the essentials of cellular life that have been perpetually exhibited for over 3.8 billion years. Indeed, the origin of life is based on the estimated age of stromatolite fossils. These represent primordial microbial mats. From this discrete evidence, it is apparent that the same self-referential cellular proclivities have reiterated forward over evolutionary space-time to continue to be expressed within all the cellular ecologies that form today’s organisms. These same cellular essentials energize and sustain natural cellular engineering and it is these self-same qualities that humans exemplify in their own successful engineering projects. 

In all circumstances, any phenotypic shift that natural cellular engineering produces is directed toward the continuous internalization of the external environment to maintain the homeorhetic equipoise of all constituent cells of the macroorganism. In this manner, biological variation represents the deliberate adjustment of cellular participants to a stream of environmental cues in the consistent attempt to maintain organismal-environmental complementarity. Importantly, collective cellular drive is always epicentered at the level of each engineering participant cell, no matter the size of the cellular collective. This is not any emergent property. Instead, it is the very basic tenet of cellular life. The cellular imperative is the protection of self-identity [7]. Collective action is its advantaged means. Thus, there is no explicit design or designer. It is always a process of self-reinforcing and iterative progression to sustain individual participants in a collective biological architecture. Its narrow aim is to permit the continuous stabilization of all constituent cells against environmental stresses. It follows that natural cellular engineering drives the holobionic life form that dominates the planet and it is in this manner that holobionts are continuously stabilized. The tissue ecologies of holobionts coordinate multiple species (eukaryotic cells and an associated microbiome) that do not have genetic commonalities. Instead, their successful association appends through an ability to share an appraisal of environmental cues through a mutualistic proteome, the reciprocal exchange of bioactive molecules and immunomodulators, and a multiplicity of shared targets. This demands a high level of cross-communication to effect coordinate engineering as an expression of on-going natural learning to support cellular homeorhesis through aggregation as collective cellular tissue ecologies. 

Every gene may be capable of polymorphic variation, but epigenetic modifications are the primary means by which organisms remain in concert with the external environment over evolutionary space-time [10]. It is argued then that given the nearly infinite complexity of holobionic life, non-random genetic utilization and editing represent living requirements, particularly due to the necessary fluidity across scales. Although phenotype is a product of cellular action, cellular action is reciprocally affected by phenotype. From within this reciprocating dynamic, natural cellular engineering, natural genetic engineering, and niche construction become sources of biological creativity balanced by cell-centered ecological constraints. When natural genetic engineering becomes the framework of biological expression, the process of evolutionary development shifts from a focus on the selection of random changes at the nucleotide level to an issue of information architecture across scales [4,152]. Its only drive is a consistent effort to maintain organismal-environmental complementarity through continuous environmental endogenization. Thus, natural cellular engineering is the active expression of the information management system of the cell. 

## 6. Discussion

Until recently, single-nucleotide polymorphisms had been supposed to comprise the feedstock of selectable variation [127]. Although many other mechanisms of genetic variation are now acknowledged, including an influential epigenome and structural genetic variations, there is still concerted resistance to assigning importance to any genetic mechanisms that could be generated from the deployment of non-random genetic events [29]. Necessarily then, evolution remains enmired within a system of adaptation premised on either replicative or induced errors. However, problematic inconsistencies in this framework are acknowledged. For example, a recent attempt has been made to reconcile the discrepancy between the standard model of a time-homogeneous genetic mutation rate and the actual pace of observable phenotypic changes [153]. Instead of a steady ratcheting, it has been proposed that mutations might occur at inhomogeneous rates even over the course of a single lifetime. The thesis is that mutations themselves beget other mutations. However, this solution yields its own obstacle. Accelerated mutations should predispose to a risk of a ‘runaway’ accumulation of mutations. A robust error correction system must therefore apply. From this, the major problem with the hypothesis surfaces. The conundrum of random genetic errors leading to fruitful biological variants cannot be solved by supposing that more frequent errors at specific ‘at-risk’ genetic segments would more conducive to evolutionary adaptation. It is thereby argued that crux of the issue of random mutation centralizes within the theory itself rather than any attempt to adjust its particulars.

Four overarching issues can be presented that contradict the theory that random mutations propel evolutionary development. The first has been long-recognized. The possibility that a concatenation of random changes could yield productive biological outcomes is recognized as improbable. This was adroitly addressed over 50 years ago by Sir Ernst Boris Chain, the winner of the Nobel Prize for Physiology or Medicine in 1971. In an address to colleagues, he noted: “To postulate, as the positivists of the end of the 19th century and their followers here have done, that the development of survival of the fittest is entirely a consequence of chance mutations, or even that nature carries out experiments by trial and error through mutations in order to create living systems better fitted to survive, seems to me a hypothesis based on no evidence and irreconcilable with the facts” [154]. No doubt, 50 years later, many who read this will bristle at its bold declaration. However, Chain has not remained alone in his contention. Many others have also acknowledged that any evolutionary narrative based on ‘error-replication is problematic [114]. It must be emphasized, however, that this stance is not intended to purport that random variations are inconsequential or infrequent. It has been pointed out that the error rate in immediate DNA sequence copying is approximately 1 in 10^4^, which translates to millions of errors in a 3 billion base pair genome [155]. However, nearly all are corrected in a complex three-stage process which requires a large number of specialized proteins and lipid membranes. This is a whole-cell engineering process that relies on cell-centered measurement. It is not the result of DNA itself since DNA outside of a cell is inert. Experimental evidence indicates that cell division is dependent on a critical charge-induced macroscopic quantum system that is centered in the cytosol of the cell [156]. Consequently, it is now suspected that it is the cytosol which permits the disorder-induced macroscopic quantum coherences that are essential to the process of cell duplication and not cellular DNA. 

There is no question that random genetic variations apply in evolution. Instead, it is an issue of context within an evolutionary entirety. Similarly, there is no doubt that there are well-developed mechanisms to control untoward genetic variances. However, productive variations do emerge in direct response to environmental stresses. It is argued that the appropriate reconciliation is to center evolution within a system that places productive mutations to deliberate use as a ‘harnessing of stochasticity’ [157]. Chance mutations count. Some are harmful such as those that are associated with genetic diseases and errors of metabolism. Others remain cryptic and do neither harm nor good. However, others can be deployed as participants in natural genomic editing and natural cellular engineering to participate in environmental adaptations. This alternative engineering model maintains that intelligent cells can purposely act to sustain their homeorhetic equipoise through their use of cellular tools, including genetic code. In any engineering project, some serendipitous events may prove useful. 

A second deficiency of random mutation theory can be readily assessed through an examination of the fossil record. Species morphologies can be extremely stable over tens of millions of years. If variation were actually due to random genetic reproductive variations that are proposed to occur at an obligatory perpetual rate, then, their steady accumulation over time would vitiate any long-term species stability. No species could continuously resist that accruing drift over long periods. However, many species remain stable for millions of years. The morphology of horseshoe crabs has remained essentially identical for 445 million years [158]. Crocodiles are virtually unchanged over 200 million years [159]. Certainly then, robust and effective mechanisms exist to prevent consequential random genetic mutations [160]. Without these corrective mechanisms, accumulated random variations over time would undermine every species-specific morphology. No species would be long-enduing. Indeed, there is a model for systems that experience any non-corrective gradual time-dependent accumulation of errors. It afflicts all computers. Any computer that is left at length without rebooting eventually accumulates errors. Programs crash. Accumulated errors do not improve complex systems. 

The third argument rests on the well-researched instances of reverse evolution in which retrograde primitive-looking forms are derived from immediate ancestors, as processes categorized as proteromorphosis and retrograde polymorphism. For example, in response to sub-lethal stress during the Permian-Triassic period, conodonts regressed to more primitive-looking forms through documented retrogradations [161]. Retrograde polymorphisms due to a suspected anoxic event have also been uncovered that affected the Cretaceous planktonic foraminiferal lineage, Ticinella-Thalmanniell [162]. A series of atavistic reversions have been found in ammonites in the Middle and Late Toarcian which has also been attributed to environmental stress [162]. Consequentially, these events demonstrate characteristics of the reversal of terminal additions that are constrained at an atavistic plateau, ‘Dollo’s point’, as the limit of the retrograde process beyond which lethal events would supervene. Indeed, evolutionary inversions are known to be frequent during extinction periods [163]. Although the concept of terminal addition remains controversial, it has recently re-considered within a framework of continuous cellular-environmental complementarity [164]. In this frame, phenotypic terminal additions accrete as environmentally-induced episodic epigenetic adjustments in patterns of cell–cell communication in fluid reciprocation with environmental cues. In such a scenario, retrograde evolution is not interdicted since, in the first instance, those additive patterns were the result of non-random cellular responses to stress built upon cell–cell signaling pathways that were actuated in response to environmental stresses [10].

The fourth, and arguably the most important deficiency in any random evolution thesis, is the foundational evidence of non-random genomic editing that is being revealed by contemporary research.

A primary tool of non-random cellular engineering is the genome. Shapiro maintains that random genetic mutations cannot account for many of the most significant evolutionary variations which generated phenotypic adaptations [115]. Instead, consequential variations are the result of complex reciprocations between cells and their ‘read-write’ genomes as natural genetic engineering. In this manner and through a number of highly integrated linkages, phenotypic adaptations are the result of complex cell-centered processes that alter the content and expression of the genome [62]. Consequently, Shapiro indicates that it is reasonable to doubt that such disparate mechanisms can function, each in their own manner, and still yield productive evolutionary outcomes [165]. 

These same intricate genomic processes have led others to question whether a random mechanism can account for productive evolutionary outcomes [114]. A range of empirical evidence has established the crucial role of non-coding DNA acting through RNA expression along complex regulatory pathways. In addition, increasing weight is being given to the contribution of regulatory RNAs from infectious events that alter genomic content [166]. These genome-invading RNA networks contradict traditional concepts of how genomes accrete heritable genetic content that represents functional biological information [167]. Many of these viral impacts are essential sources of cellular anti-viral strategies [168]. Moreover, these do not proceed in random terms within genomes. Stem-loop RNAs function as consortia that perform as competent agents in natural genome editing [169]. Infectious non-coding RNAs insert non-randomly in genomes, targeting non-coding DNA regions, and offer regulatory features rather than sourcing protein coding [114]. 

Additional evidence of non-random genome editing derives from our human engineering use of retroviral vectors for gene therapy. There is direct evidence that the integration of these retroviral vectors follows non-random patterns in mammalian systems [170]. Based on Moloney Leukemia Virus-derived vectors, this research indicates that there is a tendency for integration to occur at preferential sites. This finding has been reinforced by phylogenetic analysis indicating that the rate of genetic re-assortment during a recent Influenza A epidemic was not genomically random and only involved isolated genetic segments [171]. Further, the integration of HPV into the genome that generates cervical cancer seems to target genomic hotspots that have been localized to specific cytogenetic bands [172].

It is now known that TEs including a variety of endogenous retroviruses, retrotransposons, and DNA transposons can rewire genomic regulatory networks [142]. TEs can not only multiple themselves, but can mobilize unrelated DNA fragments which are positioned between *cis*-acting elements or other DNA locations and can become subject to re-localization [73]. Such retroelements are acknowledged participants in adaptive evolution [96] and are a source of evolutionary novelty [99,169]. Each are residues of previous infectious interchanges [92] that can affect both somatic and germ cells, contribute to cis-regulatory DNA elements, and modify transcriptional networks [108,109,173]. Notably, flexible adaptation is ongoing among the various functional and non-coding RNAs that enable RNA-mediated cell processes that contribute to the adaptive ability of cells to meet environmental impacts [174]. 

Zamai has offered a comprehensive challenge to the concept of evolution by non-random mechanisms by enumerating several non-random genome editing pathways that are stimulated by environmental stresses [116]. These are driven by a demonstrable viral-cellular symbiotic drive that leads to the viral activation of environmentally-responsive CRISPER-Cas systems, the arousal of cellular anti-viral strategies by retrotransposon-guided mutagenic enzymes, and stress-induced non-random genome editing at constrained ‘hyper-transcribed’ genetic foci. It is proposed that this latter phenomenon yields an elevated rate of intronic retrotransposon transcripts which epigenetically mark only certain areas of the genome in non-random distributions. For example, within the eukaryotic genome, there is a nonrandom and exclusive targeting of certain immunoglobulin genes. The genetic variations within the target areas may be random but the targets themselves are not. Consequently, such changes would help drive cellular adaptive responses in non-random patterns similar to immunoglobulin somatic hypermutation. 

The prior belief that viruses and sub-viral elements were merely unmitigated pathogens and parasites has been eclipsed by our contemporary viewpoint that intimate viral-cellular symbiotic associations are a dominating aspect of evolution [52,79]. Such symbiotic relationships must accommodate differing lifestyles between the participants and must reflect individualized states of preference. In that case, such associations cannot be construed as random. Research indicates that such ‘symbiotic’ virocellular sequencing relationships are centered within co-participation between cells and targeting viruses as viral symbiogenesis [175,176]. Viruses or their sub-particles are voluminous in genomes that extend across the biological spectrum from microbes to vertebrates and should therefore be construed as essential to evolutionary processes. This is not a merely theoretical construct. It is a documented cause of evolutionary novelty and adaptation. For example, a series of highly coordinated and mutualizing viral endogenization events have been detailed in the parasitoid wasp genome that have been extensively co-opted to biological expression [177].

It is now believed that only a tiny fraction of TEs are of either initial benefit or specific harm to any target organism [102]. Since this contravenes any immediate fitness effect, it has been proposed that their persistence is due to a ‘neutral balance’ between expression and regression, with eventual culling of those that are selectively harmful [91]. However, these neutral players still remain available for horizontal transfer, not merely within target genomes but between species where they are known to effect biological outcomes. Beyond their own proliferation, TEs also contribute to intergenomic mobilization and horizontal transfer of DNA segments. Notably, these are also now believed to decisively contribute to adaptation [73]. Consequently, there has been an attempt to place TEs within a larger ecological construct of an active and capacious transferable genetic mobilome that includes transposons as integrative and conjugative elements and transferable large segment DNA genomic islands [73,178]. In such a framework, it can be argued that beyond any immediate selective benefit that might be conferred by TEs, their primary purpose is their participation in a wide multi-domain mobilome as ‘on call’ feedstock for future flexible cellular responses to environmental stresses [4]. In the same manner, some indwelling long-terminal-repeat (LTR) retrotransposons can function in a fashion similar to retroviruses and can ‘infect’ other cells through horizontal transfers. One such instance is the *gypsy* element in Drosophila, known to participate in environmental adaptation [179]. 

There are advantages to placing ubiquitous TE transfers into a framework of network theory. In this context, TEs can be seen as biological agents of information transfer across the three cellular domains of life (Prokaryota, Archaea, Eukaryota) and the virome as a form of cross-communication [6]. These interchanges are not just theoretical. Recent research confirms that eukaryotic and prokaryotic cells transmit non-coding RNAs as inter-species communications via extracellular vesicles [11]. These can exert regulatory functions in target cells to effect intercellular relationships that can be either symbiotic or pathogenic. This is a major cell–cell communication pathway among holobionts and is a discrete example of non-random natural cellular and natural genetic engineering. Two essential aspects of evolutionary development have been emphasized by Witzany [160]. First, a variety of RNA agents including RNA stem-loops and mRNAs function in cellular cross-communication to assist in cellular coordination and regulation at successive cellular levels. Secondly, viruses and their subviral components can represent agents of communication. With this as a framework, any concept of random genetic errors as predominant evolutionary drivers must defer to a contemporary narrative that affirms the established role that non-random genetic content editors have within complex ‘working’ relationships among cells. These exchanges include TEs, viruses, circular DNAs, and a variety of RNA networks and stem loops that contribute genetic information, regulatory control, and immunity in sufficient commonality to actually dominate evolutionary transitions [160]. 

In opposition to the random genetic narrative, Auboeuf asserts that the prevailing gene-centric ‘top-down’ framework that originated in early Darwinism was premised on biased observations made on complex multicellular organisms. In contrast, Auboeuf argues that an alternative ‘bottom-up’ approach better fits the physicochemical properties of nucleic and amino acid polymers [180]. In that frame, the actual objective of selective fitness is meant to sustain a “bidirectional interplay between genome and phenotype that is directed towards genomic self-integrity” [180] (p. 1). In this case, phenotype is intended to preserve cellular self-identity through its confrontation with environmental stresses. 

There is evidence that this type of specific bidirectional interplay is active. Catenulida flatworms can express genes present in plasmids that are carried by symbiotic bacteria [181]. They can even express genes that are taken up by feeding. As these genetic inputs are apparently not incorporated into the flatworm genome, it can be interpreted that they represent sources of phenotypic plasticity in the face of environmental stresses that preserves underlying genomic stability. Some prions may function similarly by protecting against environmental stresses and providing a non-genomic source of phenotypic plasticity [129].

Annila and Baverstock have approached the issue of phenotype differently. They propose that genes are only a means of specifying polypeptides [182]. Those genes that serve free energy consumption contribute to cellular phenotype. Accordingly, natural selection is principally acting on mechanisms that consume energy from the environment rather than on genetic variations. In this instance, what matters most is relational information encoded within the proteome that enables phenotype [148]. Further yet, such relational information must be deemed to be physical in character since it directly contributes to the dissipation of free energy. 

It has been previously asserted that the cell should itself be considered the first example of niche construction activity [183]. Given the extent of the coordination of genetic actions within each cell, it is further argued that the genome itself should also be considered its own form of niche construction by cells in a context in which genes are tools to be deployed but still have some autonomy. Even more importantly, given the tight relationship between transcription and regulation, all genetic shifts necessarily change measurements by the cell that are utilized to maintain crucial homeorhetic equipoise. In this case, the function of the genome clarifies. It participates in cellular life as a coordinate intermediary of those informational cues that impact the cellular senome through its phenotypic exploration of the environment [7,9]. The genome is an imperative part of the overarching information management system of cells. In compressed cellular environments in which some participants have varying degrees of independence (e.g., mitochondria), some encompassing coordinating system must be functioning. It is asserted that the information management system of the cell is that overarching system. It is only this informational framework that can permit the sufficient and seamless coordination of the cellular participants that enable natural cellular engineering [4,7,9]. From this, a further conclusion devolves. Natural genetic editing is a subtext within a larger auspice of natural cellular engineering that either deploys or contains viral or subviral introgressions. 

Some of the pathways of these types of necessary constraints have been identified. PIWI-interacting RNAs (piRNAs) are known to restrain transposons [184]. In this process, long RNA molecules that originate in the cell nucleus are transported into the cytoplasm where they match to specific piRNA production sites. These sites participate in transposon silencing in animals as an intimate coordination across the nuclear envelope. 

Essential and abundant communication extends across the three cellular domains that consistently contributes to sources of variation [6]. Exosomes are bi-layer membrane-enclosed fluid compartments present in eukaryotic fluids. Exosomes are ubiquitous but can be liberated from edible plants during mammalian digestion. These exosomes can contain a variety of cellular constituents including proteins, lipids, and non-coding RNAs. Research on mammalian serum and plasma reveals that the resulting bioactive molecules can affect mammalian gene expression [185]. Even more surprisingly, plant miRNAs are involved in the regulation and transcription of mammalian target genes [186]. These effects can proceed through plant exosome products through an efflux of repercussions that induces mammalian stem cell production [187]. Though all these means, it can be directly argued that biocommunication is the critical component of variation involving extensive cross-talk among the cellular domains and the virome. Consequently, the variations that propel multicellular life should be understood as a largely non-stochastic informational interactome. 

It has been previously defended that the information management system of cells is purposed toward the acuity of its senome that integrates the reception, assessment, communication, and deployment of all of the information that it has at its disposal as a continuous arc [9,54]. Within this type of highly integrated and reciprocating narrative, the possibility that cognitive cells with their demonstrable faculties are subject to purely random genetic variations to produce adaptive responses is simply not tenable. As Goldenfeld and Woese assert, “Interventions in biological systems inevitably provoke an evolutionary response which is rapidly emerging and spatially distributed” [41] (p. 4). This attests to coordination and control. Neither of these highly reciprocating processes can be construed as merely random. Therefore, both the object of selection and the actual scope of its total impact emerges. The object of selection is the continuous maintenance of information management system of cells, either free-living or in collective form. Since the cellular information management system of cells relies on the satisfactory measurement of environmental cues, a proper reduction ensues. The object of selection is the measuring capacity of cells to meet their changing environment. Natural selection assures that the current measurements are correct, insofar as they are sufficient to meet the contemporary environment by assuring contemporaneous cellular-environmental complementarity. Pertinently, there is no need for optimization. Indeed, that would be harmful. Necessarily, cellular prediction is backwards-oriented. Its prediction must be predicated on a cognitive extrapolation of any current trends, which is also our own general means of prediction. Cells have effective memory and depend on that to deploy their resources. A highly correlative fit to one environment is much more vulnerable and easily culled in the next environmental shift. Cellular success is based on perpetual balance, as the cellular manifestation of the well-worn adage, ‘perfection is the enemy of good’. Therefore, cellular-environmental complementarity represents the continuous endogenization of environmental cues within flexible limits [4,188]. Accordingly, natural selection can now be properly appraised as a passive filter of the entire range of processes that enable information assessment, its communication, and its coordinated deployment as natural cellular engineering and cellular niche constructions [4,9]. 

If the proposition is entertained that there are consequential non-random productive influences in evolutionary development, there is a further codicil. It cannot be rare. If it were uncommon, then its effects would be vitiated by myriad subsequent random events. However, biology is cellular problem-solving to enable the continuous endogenization of the external environment [4,9,22,130,174]. In that circumstance, non-random genomic editing cannot be exceptional. In a cognitive frame, only directed cell–cell communication and measurement can underlie cellular problem-solving. Therefore, it follows that natural genomic editing is a prominent feature of living systems. 

Accordingly, exaptations, epistatic adjustments, and pleiotropisms must now be regarded as differing features of natural genetic engineering. Each of these sources of variation can be considered to be flexible genetic expressions of competent cells communicating across local and distant ecologies in continuous adaptation to environmental impositions. In each instance, their adjustments serve the cellular drive to maximize the ‘correct’ flow of biological information in collective cellular terms to meet the outward environment. In natural processes, everything depends on everything else, and phenotypes are emergent properties of their systems [182]. 

To be clear, however, none of this is meant to indicate that random variations do not occur or are inconsequential. Random occurrences can still represent solutions to environmental stresses through a ‘harnessing of stochasticity’ that can be intermittently deployed by organisms to generate new functional responses to environmental stresses [157,189]. All living entities including cells have agency by virtue of causal independence and intentionality that can exert directionality that will influence evolutionary development [190]. Through that agency, random occurrences including random genetic variances can be purposed towards discrete biological ends in the same way that a serendipitous event during a human engineering project can become a useful part of an overall outcome. 

In this manner, random variations can invigorate flexibility in the assessment of biological information and be construed as crucial tools of cells. Random events occur within genomes and affect the entire genetic complement of cells. Any of these might be recruited towards a productive end. Even the random variances of ambiguous information are not biological impediments. Our biological system is built upon obligatory uncertainties. It can be asserted that it is exactly this mandated imprecision of available information that is both the propulsive force of complex life and its source of creativity. Variances in the interpretation of uncertain data underscore all biological deployments. At every scope and scale, cognitive life is defined as the specific ability to use information to sense ambiguities and settle those uncertainties into discrete biologic action. Thus, in the context of self-referential agency, biological variations are the result of informational uncertainties which instigate responsive natural cellular engineering that produces cellular biological outcomes. Accordingly, evolutionary development becomes the perpetual entanglement of living entities in their use of information to resolve environmental ambiguities into explicate self-referential biological solutions.

Since variations are always executed at the cellular level, they are necessarily the result of coordinate measurement as natural cellular engineering in multicellular organisms. Thus, variations are themselves forms of prediction. It follows that natural cellular engineering and cellular ecological niche constructions are predictions and it further defaults that phenotype, as their product, must be too. Improved predictions entail more efficient use of cellular resources and limit the work required to sustain homeorhetic equipoise. In a cellular context, this reduces. The thrust of non-random natural cellular engineering and niche construction is the consistent collaborative multicellular effort to reach mutualized cellular states of least uncertainty [9]. That state of least uncertainty for cells can be defined. It is that homeorhetic circumstance that is consonant with the demands of the environment as preferential equipoise. Cells seek to maintain continuous organismal-environmental complementarity. Variations are their means of exploring information space to accomplish this [3,4]. Therefore, since variations are forms of conjoint prediction, they cannot be predominately random. When placed in the self-referential frame, variations are differing forms of prediction rather than chance. Phenotype is the active expression of biological variation as predictive cellular strategies to meet environmental exigencies which is dependent on the measured assessment of information by intelligent cells. Selection assures that the measurements are ‘correct’ [9].

Although most species have significant genetic variations and are subject to wide-ranging environmental influences, phenotypic variation typically proceeds within a narrow range. Robustness (canalization) is the tendency towards invariance of phenotypes and the maintenance of function despite internal and external perturbations. It is ubiquitously observed in biological systems and is regarded as a fundamental feature of evolvable complex biological systems [191]. The robustness of biological organisms and phenotypes is believed to be due to a number of factors, including gene expression and transcription, protein structure, physiological homeostasis, and metabolic drive [192]. There are also a variety of biochemical mechanisms that underlie robustness, such as thermodynamic stability at the RNA and protein levels, or the binding affinity of promoters. However, the origins of this robustness have remained a major puzzle. Towards a solution, networks of interacting transcriptional regulators have been modeled and have been shown to act to constrain the genetic system towards optimization, even absent selection [193].

The term ‘canalization’ was coined by Waddington in the 1940s to denote the ability of a population to produce the same phenotype regardless of the variability of its environment or underlying genotype [194]. Since that time, it has been viewed as a particular kind of epistasis in response to environmental stresses, such that both environmental canalization and phenotypic plasticity can be considered as dual aspects of the same phenomenon [195]. It is generally considered that canalization results in the accumulation of phenotypically cryptic genetic variations, which can be intermittently expressed under triggering conditions. Since such cryptic variation is hidden, it can be the subject of selection only when released. Flatt [195] notes that canalized genotypes maintain a cryptic potential for expressing particular phenotypes. When accumulated, such canalization may increase evolvability, at least at the population level. As canalization has been explored within a broader range through epistasis, pleiotropy, buffering, and modularity, all of which are assumed as emergent properties of gene networks, there has been an interest in identifying a unifying factor. It has been proposed that microRNAs function as those key players through their interactions with an extensive network of protein-coding genes believed to have evolved to buffer stochastic perturbations conferring robustness [196]. 

When robust systems are stressed, they adjust through positive or negative feedback loops that permit the maintenance of specific functionalities in the face of a wide variety of perturbations [191]. These mechanisms include both modularity and the decoupling of genetic variation from phenotype, providing a buffer against mutations [191,197] that likens the structure and function of these mechanisms to sophisticated engineering in view of the sheer complexity of interactions. These require autonomy at some levels and high levels of interacting feedback at others, which can be best modeled as an engineering control system that requires a coherent architecture [198]. In this manner, robustness is balanced as preservation of function against unintended fragility under uncertain and varying conditions. This overarching regulatory oversight overlaps the cellular usage of its toolkit as the cell seeks equipoise when confronted by conflicts generated by internal perturbations (mutations, fluctuations in biochemical parameters, noise) and external environmental disturbances [197]. This expresses as enduring forms of phenotypic expression and species stability across millions of years. Considering the foregoing, it can be effectively argued that all productive variations are actually the products of obligatory constraints. Constraint is essential to problem-solving, and it is now well-accepted that constraint is a requirement for evolution [7,9,22,199,200,201]. 

For many years, there has been spirited debate over the origin of biological self-organization [202,203]. However, once placed into the self-referential plane, that enduring question settles. Self-reference is self-organization [9]. When self-referential cognition is the baseline of cellular responses, then, any energetic input or received communication obliges its measured assessment. Necessarily, this expends work. That work becomes a reciprocal communicative signal to some other participant within the same biological field, either adjacent or distant. Reception of that signal initiates a repetitive cycle that constitutes a self-reinforcing and reiterating work channel. This is crucial since these types of signals can be considered to be stigmergic cues. These are forms of coordinating communication that contribute implicit order to any organisms that participate within an environment at any scale [204]. This concept has been well-demonstrated at the level of insects, for example, termites, ants, or bees. It has also been argued that this same communication nomenclature can be appropriately attributed to cells [4]. In stigmergy, any work performed by a self-referential agent leaves a trace in the environment that can stimulate further work by other self-referential agents. This represents a spontaneous and self-reinforcing form of coordinating communication and, when concatenated among cells, propels natural cellular engineering as conjoint assessment of environmental stimuli and mutualized problem-solving. This point deserves further emphasis. Natural cellular engineering is not the product of evolutionary emergence. Instead, it is a default product of thermodynamic action among self-referential cells. A necessary conclusion follows. When cells are properly placed within their self-referential cognitive frame in which ambiguous informational inputs must be measured, natural cellular engineering is an ineluctable result. Importantly, natural cellular engineering is not seeking any solutions in the sense of any specific macroorganic outcomes. Those are emergent phenomena of cellular problem-solving whose narrow aim it to assure that each individual cellular participant at any level of organization structure is permitted to sustain homeorhetic equipoise. Multicellularity exists because each cellular participant can best maintain itself through collective measurement that improves the validity of environmental information. Clearly then, natural cellular engineering is ‘natural’ insofar as it is a direct action-reaction to basic thermodynamic forces in a self-referential frame. Across evolutionary space-time, there is a single consonant bioactive direction: the maintenance of self-referential cellular homeorhetic equipoise. Genes and all other intracellular structures, such as the variety of cellular membranes, are tools. The result is the collective, competitive and collaborative assessment of information directed towards cellular solutions to cellular problems. As Stuart Kauffman has aptly stated, living things can be defined by their ability to “act on their own behalf” [205] (p. 73) in indeterminate circumstances. 

In such circumstances, variation is not an accidental effect, but a manifestation of collective cellular problem-solving. New environmental challenges demand variation. Since biology is energetically parsimonious, variations are a cellular coping strategy to deal with environmental stresses through efficient means [206]. Necessarily, variants assess information differently. It is that differential assessment that offers its potential for a more advantaged defense of the self-referential states of each of the constituent cells. This is the point of natural cellular engineering as a conjoined confrontation with any new unfolding and agitating environment. It follows that any variant adaptation is a form of strategic information management. Its direction is the maximization of the soundness of environmental cues in the maintenance of cellular self-integrity. As noted, this can be defined as an organismal state of least uncertainty from which advantaged predictions might ensue. This reinforces that in the context of an information management system for holobionts in their reactions to environmental cues, neither a top-down nor bottom-up narrative suffices. In complex multicellular organisms that are deeply interconnected among a vast number of participants that all interact in a complex manner with the environment, there can be no ‘privileged level of causation’ [207]. 

Once this is perceived, the reason why the concept of evolution through random genetic variations must be abandoned disentangles. Random genetic variations represent a source of uncertainty for living organisms. This explains why multiple mechanisms to police against genetic errors are commonplace. If random variations were a boon to cells as a productive exploration of information space, then they would not be vigorously constrained. From this, the purposes of variant engineering clarify. Variation is the cellular strategy for environmental exploration directed as life’s permanent requisite: the consistent successful internalization of the external environment [3,4,9,151,188]. Cellular life meets that requirement through self-referential cell-based measurement of environmental cues at two concurring levels. All cellular measurements are enacted at the individual cellular level. Through cell–cell communication, cellular measurements aggregate as collective measuring environmental assessment. The manner in which these summated measurements deploy is natural cellular engineering, reliant on non-random natural genetic editing to provide heritable active cellular solutions to environmental stresses. 

## 7. Conclusions

All the essentials of multicellular life have emerged from unicellular particulars. Every aspect of present life has extended forward across billions of years from a distinguishable congruent point of initiation. Every cell has individual self-referential awareness that permits the assessment, deployment, and communication of information for contingent problem-solving. From within that base, Cognition-Based Evolution represents a consonant evolutionary alternative to the conventional contention that evolution is governed by random genetic variations. Instead, evolutionary development is energized by intelligent cells upholding their own fates through the measuring assessment of information and its deliberate communication. Consequently, cellular actions are based within measurement. It defaults that this represents cellular prediction. Therefore, it cannot be random. To enhance the validity of their measurements and improve their predictions, cells enter into multicellular collaborative associations. This enacts the multicellular ecologies that yield biofilms or holobionts. All of these are the product of the measured assessment of biological information among their varied participants. Collectively, this productive use of cellular information energizes as natural cellular engineering and mutualizing niche constructions. As these latter conjoin, phenotype emerges within biofilms or holobionts to cope with environmental stresses through the exploration of the outward environment and the acquisition of epigenetic information. As all such outcomes are dependent on the assessment of information, natural cellular engineering represents the bioactive expression of the cellular information management system. Through this means, collaborative cells integrate environmental cues to regulate life-cycle biological development and provide the propulsive thrust of evolutionary variation.

It has been previously asserted that every novel function derives from a pre-existing one [208]. However, how genetic variations actually become biological creativity had remained a mystery. It is maintained that the origins of biological variation in holobionts should be explored through the dimension of intelligent measuring cells. Through this means, varying patterns of environmentally responsive linked cellular ecologies are deployed to effect the continuous internalization of the outward environment in the constant defense of cellular self-integrity. Among their many threats, the cellular defense against intruding pathogens is paramount. In such circumstances, genes are tools of cells in engineering robust immunity directed towards cellular self-integrity. That intricate process and its ensuing variations are the result of predominately non-random natural genomic editing and engineering. Certainly though, random genetic variations do supervene. These may be silenced or deployed as cell-centered ‘harnessing of stochasticity’ to provide further engineering solutions to stress. Consequently, selection can now be assessed as an over-arching post-facto filter of antecedent natural cellular engineering solutions to cellular problems. Selection imposes continuous cellular-organismal-environmental complementarity in confrontation with environmental stresses and assures that consensual cellular measurements meet current environmental stresses with sufficient flexibility. All living forms attest to that success.

From these rudimentary beginnings, all life on the planet spills forward in its myriad variations. No matter the means by which genetic variations arise, their entire panoply is actively commissioned toward self-directed cellular problem-solving, perpetually maintaining the tenuous balance of symbiotic and competitive relationships across the cellular domains and the virome.

## Figures and Tables

**Figure 1 cells-10-01125-f001:**
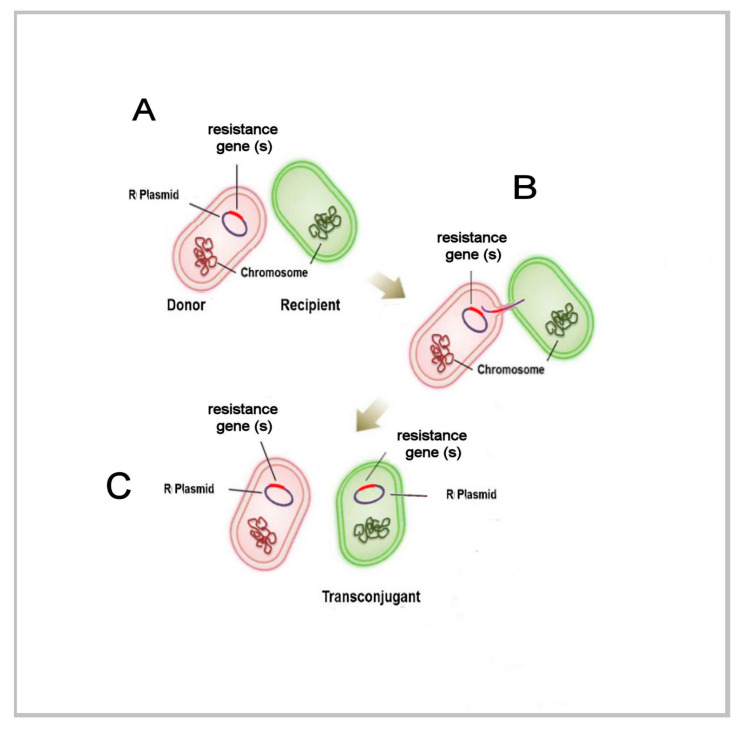
Antibiotic-resistant plasmid conjugative transfer in prokaryotes. (**A**) In biofilms, via complex cell–cell communication mechanisms believed to be activated through quorum sensing, potential donor and recipient cells identify one another for the possible transfer of antibiotic-resistance genes. (**B**) A donor cell with plasmid-based antibiotic resistance genes will attach to a recipient cell via a narrow pilus, a hair-like appendage as a conjugative bridge to effect resistant gene (R plasmid) transfer [42]. (**C**) After transfer, both the donor cell and the tranconjugant recipient are equipped with an R plasmid antibiotic-resistant gene(s).

**Figure 2 cells-10-01125-f002:**
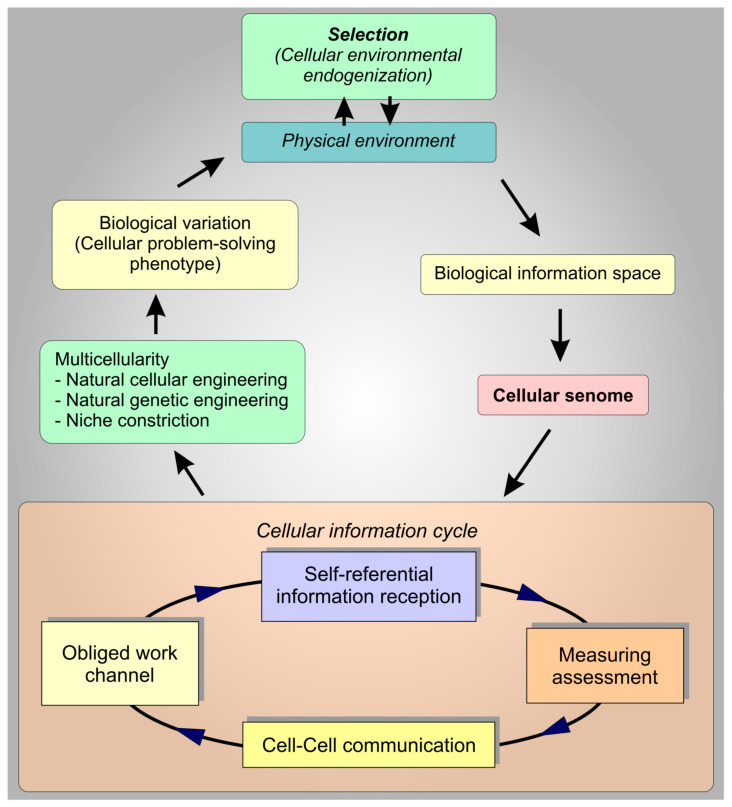
An informational interactome governs natural cellular engineering.

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
