# Peer review of "Non-Random Genome Editing and Natural Cellular Engineering in Cognition-Based Evolution"

_cells, 2021, doi:10.3390/cells10051125_

Round 1

Reviewer 1 Report

This manuscript supports the theory of the "Cognition-Based Evolution". To be honest I was not aware of such a theory, which try to explain many known biological processes from a non-genome centric view. 
I will not comment on the theory itself (which hasn't captured my interest). Instead I will try to focus on the many mistakes, misinterpreted citations and hard to follow passages in the text. 
In my opinion there are several of the authors' comments on the cited literature which are "forcedly" interpreted to fit the above mentioned theory. In addition the manuscript is too long and this can make the readers losing interest to reach the end of the text.

Please find below my comments on the manuscript.

Figure 1. A figure legend with the full explanation of each element in the picture should be provided. Also, in the trans-conjugant the "R plasmid" should be indicated 

Quorum sensing is mentioned in the text but it is never referenced. Please provide reference and explain briefly why QS is important for.

I have never heard that bacteria exchange DNA for nutrients through the conjugative pilus. In my opinion this concept is wrong. Please, revise this part carefully

p4 l166 "two-way transfer" transfer of what? While it is clear that the donor cell send a DNA molecule to the recipient cell, what it is transferred in the opposite direction is not. Please explain and provide references.

§p4 l170 "a donor prokaryote with antibiotic resistance genes will attach to an antibiotic-susceptible prokaryote". Both the donor and the recipient could be resistant (or both susceptible) to an antibiotic, which is not necessarily the transferred one. Simply refer to donor cell and recipient cell.

p7 l294 "source of new genetic code" referring to genetic code could be misunderstood by the reader. Try using different terms (e.g. variation)
paragraph 4 subpara 4.1

p7 l299 "it has been proposed that transposons may be the origin of those viruses that have contributed many of the fundamental properties of living organisms [56]" This reference is not appropriate. 

p8 last two sentences are not supported by the given reference ( 73)

p9 l392 could you please replace stimulate with another word? 

p9 l395 which type of exchange the authors are referring to?

p9 l420-424 the fact that conjugation is a highly organized horizontal transfer event does not necessarily means that all HGT events should be similarly coordinated. We basically know anything about the HGT process as it occurs in nature. The Authors should therefore try to describe the current literature without overstating conclusions, only to support their theory.

p10 l438 "I" is missing at the beginning of the sentence

p10 l450 "Escehrichiaw" please remove the final w

p10 l475-479 I would suggest adding a couple of references to recent reviews describing the  contribution of TEs to the host regulatory sequences in mammals (Sundaram et al., 2020 (doi:10.1098/rstb.2019.0347)) and in model organisms (Moschetti et al., 2020 (doi: 10.3390/biology9020025))

p11 l530-533 Not clear. Please revise these sentences

p11 l542 stem-loop is not a "category" of structural variation. Maybe a source?

p11 l543 "genetic strand" replace with "nucleic acid"

p11 l544 "hair-pin loop" please put dash in the correct position

p12 l546 "as signals in protein sequencing" Are you sure? Please provide an explanation or revise

p19 l883 rather than a single reference at the end of the sentence I would put some additional references to recent papers after each of the points mentioned, I would suggest   the same references suggested above (Sundaram et al., 2020 (doi:10.1098/rstb.2019.0347)) and Moschetti et al., 2020 (doi: 10.3390/biology9020025)) to justify "cis-regulatory DNA elements"

reference 139 has some extra text (Presidential Address)

Author Response

Responses to Reviewer 1:

The co-authors are deeply grateful for the excellent suggestions of the reviewer. We were not surprised to learn that reviewer was unfamiliar with the framework of Cognition-Based Evolution, although it is a concept that has been thoroughly peer-reviewed. Indeed, this is the specific reason why the manuscript was submitted for this Special Issue after Dr. Zamai's kind invitation.  The co-authors believed that the readership of Cells would be interested in knowing that an alternative evolutionary narrative exists that is centered within the unfamiliar framework of cellular cognition. The co-authors recognize that the length of the manuscript might be a barrier to some readers. However, in our opinion, the length is necessary to vigorously defend an unconventional framework. We thank the reviewer for permitting this opportunity to introduce the concepts of Cognition-Based Evolution to the discerning readership of Cells.

Please see our responses below to each criticism in brackets with the preceding questions in italics. [Note bene: the lines referenced below pertain to the Word format required for the Cells re-submission.]

Figure 1. A figure legend with the full explanation of each element in the picture should be provided. Also, in the trans-conjugant the "R plasmid" should be indicated.

[ Fig. 1 has been amended as requested. Letter labels have been added for each stage. The figure legend has been amended to reflect the R plasmid in the transconjugant as requested. The statement about 'nutrient exchange' has been deleted. ]

Quorum sensing is mentioned in the text but it is never referenced. Please provide reference and explain briefly why QS is important for.

[Amended to inidicate what quorum sensing is among cells and Ref. 39 was added.  Please see lines 183-186.]

I have never heard that bacteria exchange DNA for nutrients through the conjugative pilus. In my opinion this concept is wrong. Please, revise this part carefully.

[The co-authors have carefully examined their notes and source materials. On review, we agree with the reviewer that there is insufficient evidence to make the statement that plasmid exchange is reciprocated by a nutrient exchange from the recipient. Although it is now known that intercellular nanotubes can function in the exchange of metabolic nutrients, (please see-  Pande, S., Shitut, S., Freund, L. et al. Metabolic cross-feeding via intercellular nanotubes among bacteria. Nat Commun 6, 6238 (2015). https://doi.org/10.1038/ncomms72380), we agree that this does not, in itself, support the assertion that such a transfer is occurring during plasmid transfer. Thank you for this opportunity to correct this error.]

p4 l166 "two-way transfer" transfer of what? While it is clear that the donor cell send a DNA molecule to the recipient cell, what it is transferred in the opposite direction is not. Please explain and provide references.

[Amended and corrected by deleting 'two-way transfer' and more carefully indicating that the exchange is from donor to recipient cells, lines 189-190.]

  • p4 l170 "a donor prokaryote with antibiotic resistance genes will attach to an antibiotic-susceptible prokaryote". Both the donor and the recipient could be resistant (or both susceptible) to an antibiotic, which is not necessarily the transferred one. Simply refer to donor cell and recipient cell.

[Amended in the text and in the legend to Fig. 1. to more clearly indicate that the transfer is from donor to recipient cells. Thank you.]

p7 l294 "source of new genetic code" referring to genetic code could be misunderstood by the reader. Try using different terms (e.g. variation)
paragraph 4 subpara 4.1

[Amended, lines 345-346, changing 'genetic code to 'new genetic variations'].

p7 l299 "it has been proposed that transposons may be the origin of those viruses that have contributed many of the fundamental properties of living organisms [56]" This reference is not appropriate.

[Multiple mistakes within the sentence were corrected, now indicating that transposons can be the source of regulatory networks which is supported by the reference, lines 350-353.   Thank you.]

p8 last two sentences are not supported by the given reference ( 73)

[Amended to indicate that some microbial associations may participate and by deletion of the following sentence,  lines 435-438. The remaining assertion is supported by the reference. Thank you.]

p9 l392 could you please replace stimulate with another word?

[Amended, substituting 'participate',  line 449.]

p9 l395 which type of exchange the authors are referring to?

[Amended to indicate 'retrotransposition events', line 452].

p9 l420-424 the fact that conjugation is a highly organized horizontal transfer event does not necessarily means that all HGT events should be similarly coordinated. We basically know anything about the HGT process as it occurs in nature. The Authors should therefore try to describe the current literature without overstating conclusions, only to support their theory.

[Substantial amendments have made across the paragraph less assertively state its points, lines 479-484.]

p10 l438 "I" is missing at the beginning of the sentence

[Amended, thank you.]

p10 l450 "Escehrichiaw" please remove the final w

[Amended].

p10 l475-479 I would suggest adding a couple of references to recent reviews describing the  contribution of TEs to the host regulatory sequences in mammals (Sundaram et al., 2020 (doi:10.1098/rstb.2019.0347)) and in model organisms (Moschetti et al., 2020 (doi: 10.3390/biology9020025))

[Amended with the addition of both references, thank you, lines 546-551].

p11 l530-533 Not clear. Please revise these sentences

[Amended through clarification of these sentences, lines 602-608].

p11 l542 stem-loop is not a "category" of structural variation. Maybe a source?

[Amended and changed 'category'  to 'source', line 616]

p11 l543 "genetic strand" replace with "nucleic acid"

[Amended, changing 'genetic strand' to 'nucleic acid sequences', line 618].

p11 l544 "hair-pin loop" please put dash in the correct position

[Amended].

p12 l546 "as signals in protein sequencing" Are you sure? Please provide an explanation or revise

[Amended changing 'signals of protein sequencing to 'building blocks of  ribozymes and messenger-RNAs' lines 620-622]. 

p19 l883 rather than a single reference at the end of the sentence I would put some additional references to recent papers after each of the points mentioned, I would suggest   the same references suggested above (Sundaram et al., 2020 (doi:10.1098/rstb.2019.0347)) and Moschetti et al., 2020 (doi: 10.3390/biology9020025)) to justify "cis-regulatory DNA elements"

[Amended adding the references, 'Each are residues of previous infectious interchanges [92] that can affect both somatic and germ cells,  contribute to cis-regulatory DNA elements, and modify transcriptional networks [108,109, 173], lines 992-994.  Thank you].

reference 139 has some extra text (Presidential Address)

[Amended].

Reviewer 2 Report

The manuscript by Miller, Enguita and Leitão, continues to develop the concept of CBE (Cognition-Based Evolution), this time in the context of non-random genetic engineering. The manuscript is fairly long, but it is generally well-written and well-supported by the appropriate evidence. I am therefore, sympathetic to the views expressed by the authors and do not have any major objections. However, there are some areas that the authors need to clarify or explain better. Some of those areas are highlighted below.

The concept of homeostasis. On several occasions the term “homeostasis”, and its variations such as “homeostatic equipoise”, are used in the text, mostly to defend the idea that the cellular homeostasis is the constant around which various physiological parameters are maintained. The term homeostasis means steady state and it is usually associated with Claude Bernard, even though the term was popularised in English by Walter Bradford Cannon. Homeostasis, when used in the evolutionary context, implies, to a degree, the lack of dynamical change. In other words, if there is one state of cellular/organismal physiology that needs to be maintained then major changes would not be possible. In other words, only minor changes are permitted – maintenance of parameters around fixed regulatory setpoints (e.g. temperature etc,). However, in the context of evolution everything is in the flux – there is no single state of the system. If a steady state is to be maintained there would be no new species, or new cognitive forms. This is particularly important from the perspective of CBE. Therefore, a term more appropriate than homeostasis, in the evolutionary context, would be its variation known as homeorhesis, which means a steady flow. The term was coined by Conrad Hal Waddington, in the context of animal development. I suggest that the authors consider using the term homeorhesis in the evolutionary context and restrict to the term homeostasis in the physiological context.  

“Interactome”, lines 90-91 – this term is used in a similar context by Slijepcevic. See Journal for General Philosophy of Science Principles of information processing and natural learning in biological systemshttps://link.springer.com/article/10.1007/s10838-019-09471-9. I suggest the authors acknowledge the use of the term “interactome” by others.   

“By unknown means”, lines 55-56 – this statement may be interpreted differently. Cognition is a form of learning. Any learner will not have adequate knowledge at the beginning of the process, but will at the end. So the expression “by unknown means” may be interpreted as the product of the learning procerss.

Lines 134-149, CBE is seems to be exclusively based on informatics. However, synergism and symbiosis may have an important role to play in cognition processes as explained by other authors. It may be worth identifying the links, with for example Maturana and Varela concept of autopoiesis and Fritjof Capra interpretation of cognition. In addition, given that the concept of information and cellular measurement are used extensively in CBE it is curious that the originator of the concept of biological information, Gregory Bateson, is not mentioned. In particular how he viewed biological information

Lines 618-619, Is the information physical in biology? Not according to Gregory Bateson or Peter Corning. Also, information does not represent only the transfer of energy/. It has other forms, in particular in the context of relational biology.

Line 689-691 If transfer of energy is information this has some implications for the metabolism first hypothesis when interpreting the origin of life (line 705). This means that not only genes are cellular tools but also the metabolic information. The authors should consider elaborating on different interpretatiosn of origin of life in context of CBE: gene first hypothesis (RNA world) and metabolism first hypothesis (e.g. autocatlytic sets, homeostasis etc. as argued by Stuart Kauffman and others.

Line 714 – terms biological variation in the evolutionary sense and homeostasis are not compatible. For this reason the term homeorhesis may be better (see my earlier comment).

The term engineering has a peculiar meaning in the context of CBE. It seems to me that CBE is an ecological rather than a cybernetic concept. In ecology mind is more than computation. How do you integrate mind with CBE? Worth consulting Bateson’s book Mind and N`ture.

Lines 730-731, it may be appropriate to include biosemiotics in the context of cross-kingdom communication.

The term problem-solving is cybernetic rather than ecological. It may be better to use the term natural learning. Learning is not only about problem solving, but placing the newly acquired knowledge in a wider context which is a form of ecological epistemology. The concept of holobionts is a good example.

Lines 866-876 – there is a range of paper in this regard which indicate that the genome organistion in terms of gene distribution or chromatin organisation is not random. This means that if the genomics is viewed as a form of natural engineering than it is highly non-random. The process leaves traces that are visible across the species range in the forms of chromosome band organisation. In that sense the concepts of gentotype and phenotype are complemented by the concept of the nucleotype which reveals the extent of natural engineering. The nucleotype concept introduces a new level of biological hierarchy and facilitates inclusion of factors ignored by the classical genotype and phenotype hierarchies. The new factors include: nuclear structure and cell size, cell division times and developmental rates, chromosome territories within the interphase nuclei, 3 D nuclear topology, mechanical forces acting on the cell, selfish gene level selection, supraorganismal level selection such as group selection and genetic drift.

Line 901 – It is curious that the term symbiosis is mentioned only in the context of cellular-viral interactions. An example of natural engineering is endosymbiosis – archeal-bacterial merger – as the greatest discontinuity in the history of life also known as eukaryogenesis. Also, the concept of holobiont is elaborated by the main proponent of symbiosis as the driver of evolution, Lynn Margulis. It may be worth trying to reconcile endosymbiosis with CBE.    

Lines 910 and earlier – estimates suggest that 45% of the human genome is retroviral in origin. See for example Moelling and Broecker 2019.

Line 955 – the term genomic stability is strictly speaking applicable only to species that reproduce sexually. Bacteria and archaea are genomically unstable. The HGT is widespread there. Shapiro’s “read-write genome” is a testament to the bacterial genomic freedom. Some microbiologists (for example Sorin Sonea) argued that bacteria do not have species. Genome stability maintenancd may be more important to eukarya. The stable genome is a late evolutionary event. It emerged only after >2 billion years of evolution. Current estimates suggest that eukaryogenesis can be traced to between 1 – 1.6 billion years from now.

Lines 1038 – this is well stated. It may be worth citing the thought that natural selection is an editor but not the creator. The term creator here is meant as the originator of evolutionary novelty by the action of autopoietic cognitive agents (as opposed to intelligent design).

Line 1045 - However, biology is cellular problem-solving – this is difficult to justify. The Gaia theory suggests that cells are integrated with the higher hierarchical level.

Lines 1144-45 – constraint. It is curious that neodarwinists are cited in the context of constraints. I suggest that additional references about constraints are included that are more in line with the present context.

Line 1182 etc - In such circumstance, variation is not an accidental effect, but a manifestation of collective cellular problem-solving. New environmental challenges demand variation. Since biology is energetically parsimonious, variations are a cellular coping strategy to deal with environmental stresses through efficient means

In the evolutionary context variation is not consistent with homeostasis and with cells. There are top-down factors that influence variation including symbiosis and the biosphere as explained in the context of Gaia theory. CBE, as presented, seems to be a a bottom up view. But the top-down view should also be taken into account.

My final comment: given that the paper is long, it was difficult to register all typos or textual inconsistencies. I recommend that the authors thoroughly check the manuscript to minimise textual inconsistencies.

Author Response

Response to Reviewer 2.

The co-authors are very grateful for the positive review of our manuscript and thank the reviewer for such a constructive and valued critique.  Further, the co-authors believe that the manuscript has been  strengthened based opon the reviewer's comments. Please see our responses below to each criticism in brackets with the questions in italics. [Note bene: the lines referenced below pertain to the Word format required for the Cells re-submission.]

The manuscript by Miller, Enguita and Leitão, continues to develop the concept of CBE (Cognition-Based Evolution), this time in the context of non-random genetic engineering. The manuscript is fairly long, but it is generally well-written and well-supported by the appropriate evidence. I am therefore, sympathetic to the views expressed by the authors and do not have any major objections. However, there are some areas that the authors need to clarify or explain better. Some of those areas are highlighted below.

The concept of homeostasis. On several occasions the term “homeostasis”, and its variations such as “homeostatic equipoise”, are used in the text, mostly to defend the idea that the cellular homeostasis is the constant around which various physiological parameters are maintained. The term homeostasis means steady state and it is usually associated with Claude Bernard, even though the term was popularised in English by Walter Bradford Cannon. Homeostasis, when used in the evolutionary context, implies, to a degree, the lack of dynamical change. In other words, if there is one state of cellular/organismal physiology that needs to be maintained then major changes would not be possible. In other words, only minor changes are permitted – maintenance of parameters around fixed regulatory setpoints (e.g. temperature etc,). However, in the context of evolution everything is in the flux – there is no single state of the system. If a steady state is to be maintained there would be no new species, or new cognitive forms. This is particularly important from the perspective of CBE. Therefore, a term more appropriate than homeostasis, in the evolutionary context, would be its variation known as homeorhesis, which means a steady flow. The term was coined by Conrad Hal Waddington, in the context of animal development. I suggest that the authors consider using the term homeorhesis in the evolutionary context and restrict to the term homeostasis in the physiological context.

[The co-authors particularly value this comment. It is an excellent suggestion and is entirely enlightening. Many thanks. The manuscript has been accordingly amended throughout with some introductory explanation in the abstract.]

Interactome”, lines 90-91 – this term is used in a similar context by Slijepcevic. See Journal for General Philosophy of Science Principles of information processing and natural learning in biological systemshttps://link.springer.com/article/10.1007/s10838-019-09471-9. I suggest the authors acknowledge the use of the term “interactome” by others.

[The co-authors agree that some direct acknowledgment of antecedents for the concept of an informational interactome is necessary and an amendment is has been gratefully made with an addition of this reference [26],  lines 104-108.]

“By unknown means”, lines 55-56 – this statement may be interpreted differently. Cognition is a form of learning. Any learner will not have adequate knowledge at the beginning of the process, but will at the end. So the expression “by unknown means” may be interpreted as the product of the learning process.

[ Thank you. An amendment has been made by deletion, line 61. Just to note, Slijepcevic, P. Natural Intelligence and Anthropic Reasoning. Biosemiotics 2020, 13, 285-307, doi:10.1007/s12304-020-09388-7., was already included as reference #15.].

Lines 134-149, CBE is seems to be exclusively based on informatics. However, synergism and symbiosis may have an important role to play in cognition processes as explained by other authors. It may be worth identifying the links, with for example Maturana and Varela concept of autopoiesis and Fritjof Capra interpretation of cognition. In addition, given that the concept of information and cellular measurement are used extensively in CBE it is curious that the originator of the concept of biological information, Gregory Bateson, is not mentioned. In particular how he viewed biological information

[ The co-authors agree and are very familiar with each of these scientists and their ideas. Indeed, their important contributions have been discussed in prior papers on CBE that were focused on information. The co-authors beg the indulgence of the reviewer in this instance. This manuscript has already been criticized as being too long and there is concern that this topic, though of vital interest to the co-authors, is not an epicenter of interest for the readership of this journal.]

Lines 618-619, Is the information physical in biology? Not according to Gregory Bateson or Peter Corning. Also, information does not represent only the transfer of energy/. It has other forms, in particular in the context of relational biology.

[The co-authors agree. There are arguments to be made on either side of this issue, e.g. Baverstock in favor of physicality, etc. Our more specific aim was to point out that information 'achieves' physicality within biological systems, thus leading into the problems of information degradation. An amendment to clarify this has been made.]

Line 689-691 If transfer of energy is information this has some implications for the metabolism first hypothesis when interpreting the origin of life (line 705). This means that not only genes are cellular tools but also the metabolic information. The authors should consider elaborating on different interpretatiosn of origin of life in context of CBE: gene first hypothesis (RNA world) and metabolism first hypothesis (e.g. autocatlytic sets, homeostasis etc. as argued by Stuart Kauffman and others.

[ This is a topic of considerable interest to the co-authors. It would be a delight to delve into it. However, the manuscript purposefully avoids this fascinating topic as beyond its scope. Given the vast compendium of opinions, the co-authors chose to skirt the issue, admitting that we have no idea how cognition began, but insisting that it did and hence, represents the correct framework into which biology should be placed. Additionally, the co-authors are concerned that the length of the manuscript necessitates certain unfortunate limitations as it is.]

Line 714 – terms biological variation in the evolutionary sense and homeostasis are not compatible. For this reason the term homeorhesis may be better (see my earlier comment).

[Agreed and many thanks for pointing this out. Amendments have been made throughout the manuscript changing most instances of homeostasis to homeorhesis ]

The term engineering has a peculiar meaning in the context of CBE. It seems to me that CBE is an ecological rather than a cybernetic concept. In ecology mind is more than computation. How do you integrate mind with CBE? Worth consulting Bateson’s book Mind and N`ture.

[ This is a highly complex and fascinating topic. The co-authors are familiar with the literature and have made the difficult decision to eschew this issue in the interests of focusing on what the Cells reader base is likely to find of greatest value to them. For this manuscript, the sense of cellular intelligence that is trying to be conveyed is meant to be equated with the general concept of being able to 'uphold oneself' (Kauffman or the excellent cited articles by Ford). The complex arguments that would be necessary to defend 'mind' at the cellular level have been extensively made, most recently by Reber in The First Minds: Caterpillers, Karyotes, and Consciousness, but in candor, would take an entire fresh article to properly articulate. In the judgment of the co-authors, this merits extensive discussion that is outside the scope of this particular manuscript, which is already quite complex.]

Lines 730-731, it may be appropriate to include biosemiotics in the context of cross-kingdom communication.

The term problem-solving is cybernetic rather than ecological. It may be better to use the term natural learning. Learning is not only about problem solving, but placing the newly acquired knowledge in a wider context which is a form of ecological epistemology. The concept of holobionts is a good example.

[Amended, lines 826-828. The co-authors concur that the term 'natural learning' is an excellent means of expressing the the manner in which cross-communication enables problem-solving at the cellular level.]

Lines 866-876 – there is a range of paper in this regard which indicate that the genome organistion in terms of gene distribution or chromatin organisation is not random. This means that if the genomics is viewed as a form of natural engineering than it is highly non-random. The process leaves traces that are visible across the species range in the forms of chromosome band organisation. In that sense the concepts of gentotype and phenotype are complemented by the concept of the nucleotype which reveals the extent of natural engineering. The nucleotype concept introduces a new level of biological hierarchy and facilitates inclusion of factors ignored by the classical genotype and phenotype hierarchies. The new factors include: nuclear structure and cell size, cell division times and developmental rates, chromosome territories within the interphase nuclei, 3 D nuclear topology, mechanical forces acting on the cell, selfish gene level selection, supraorganismal level selection such as group selection and genetic drift.

[The co-authors are deeply grateful to the reviewer for including this information in the review. We were unfamiliar with this powerful concept. We did include some discussion of chromosomal and RNA structural motifs in this paper and and will very definitely be incorporating these additional factors into future papers. Many thanks, indeed!]

Line 901 – It is curious that the term symbiosis is mentioned only in the context of cellular-viral interactions. An example of natural engineering is endosymbiosis – archeal-bacterial merger – as the greatest discontinuity in the history of life also known as eukaryogenesis. Also, the concept of holobiont is elaborated by the main proponent of symbiosis as the driver of evolution, Lynn Margulis. It may be worth trying to reconcile endosymbiosis with CBE.

Lines 910 and earlier – estimates suggest that 45% of the human genome is retroviral in origin. See for example Moelling and Broecker 2019.

[The co-authors are in complete agreement with the reviewer. Endosymbiosis as the origin of Eukaryota is noted and cited in the text in section 4.2. While the reviewer is correct in noting that symbiosis as a driver in holobionic life is not specifically addressed, our work emphasizes the paramount nature of this driver in other publications. This manuscript is shaped to address the more narrow and unfamiliar scope of non-random genome editing, which is primarily the result of viral-cellular interactions, hence our discussion of that aspect of symbiosis rather than others. Our deliberate intent of using symbiosis for this set of interactions is meant lead into the topic of viral symbiogenesis which might be generally unfamiliar and thereby informative for many readers]

Line 955 – the term genomic stability is strictly speaking applicable only to species that reproduce sexually. Bacteria and archaea are genomically unstable. The HGT is widespread there. Shapiro’s “read-write genome” is a testament to the bacterial genomic freedom. Some microbiologists (for example Sorin Sonea) argued that bacteria do not have species. Genome stability maintenanced may be more important to eukarya. The stable genome is a late evolutionary event. It emerged only after >2 billion years of evolution. Current estimates suggest that eukaryogenesis can be traced to between 1 – 1.6 billion years from now.

[Yes, in complete agreement. Amended changing 'genomic stability' to 'cellular self-integrity',  lines 1076-1078.]

Lines 1038 – this is well stated. It may be worth citing the thought that natural selection is an editor but not the creator. The term creator here is meant as the originator of evolutionary novelty by the action of autopoietic cognitive agents (as opposed to intelligent design).

[Thank you. Yes, in complete agreement. Our choice was to use the term 'filter' since it denotes that the environment imposes restrictions but makes no decisions. We were concerned that the term 'editor' might carry a hint of agency to some readers and reviewers.]

Line 1045 - However, biology is cellular problem-solving – this is difficult to justify. The Gaia theory suggests that cells are integrated with the higher hierarchical level.

[The reviewer's point is well-taken. An amendment is offered to be in the spirit of Gaia, i.e. 'cellular problem-solving to enable the continuous endogenization of the external environment', lines, 1172-1173]

Lines 1144-45 – constraint. It is curious that neodarwinists are cited in the context of constraints. I suggest that additional references about constraints are included that are more in line with the present context.

[Amended with the addition of references, line 1285]

Line 1182 etc - In such circumstance, variation is not an accidental effect, but a manifestation of collective cellular problem-solving. New environmental challenges demand variation. Since biology is energetically parsimonious, variations are a cellular coping strategy to deal with environmental stresses through efficient means

In the evolutionary context variation is not consistent with homeostasis and with cells. There are top-down factors that influence variation including symbiosis and the biosphere as explained in the context of Gaia theory. CBE, as presented, seems to be a a bottom up view. But the top-down view should also be taken into account.

[ Amended, thank you. A clarification emphasizes the bi-directionality of causation indicating that there is no privileged level and adding a pertinent reference [207],  lines 1337-1342].

My final comment: given that the paper is long, it was difficult to register all typos or textual inconsistencies. I recommend that the authors thoroughly check the manuscript.

[Agreed and the re-submission has been meticulously searched for errors. Thank you.]

Reviewer 3 Report

Miller et al. Non-random Genome Editing and Natural Cellular Engineering in Cognition-Based Evolution
1. AA. defend the sensible view that evolution is not just a history of changes in allelic or genotypic frequencies, or a history of competition to the exclusion of other kinds of interactions, including mutually advantageous ones. In this respect, this is in agreement with a variety of criticism of reductionist and strictly gene-centric Neodarwinism – a criticism expressed in the last decades by many authors under different umbrella terms, such as Extended Evolutionary Synthesis (EES), Niche Construction Theory (NCT) and others, as listed also in this ms., but not discussed.
2. Similar to other views that take more or less explicitly distance from Neodarwinism, AA. seem to deny, or at least strongly downplay, the role of selection. Selection, however, could easily explain stories of adaptation that the AA. would explain otherwise. Why not selection? E.g.:
Lines 256-267: “Collective biofilms represent a predominant form of prokaryotic life since they permit collective engineering as effective form of cellular problem-solving. In Bacillus subtilis biofilms, the participating cells can sub-specialize to elaborate an optimal extracellular matrix for either cell surface adhesion or colony mobility [48]. The specific composition of this extracellular matrix must differ to meet exact environmental and surface conditions to permit either adhesion or motility. In order to produce both, constituent cells partner in specialized activities that completely differ from those that they exhibit within their free living form [49]. These are consensual actions. There are high levels traded resources and some participants accept the voluntary loss of some cellular functions to support the collective whole [50,51]. Each of these united cellular forms with their exclusive and different outputs represents a type of phenotype and each variety has its differential discrete function, whether to stick or to migrate.
In this case, and more generally, the critical issue is not selection, but the origin of selectable variation. Selection, for example, can operate on partnerships (between cells of the same or different kinds) with different fitness in a given environment. Same as for any biological systems considered from the point of view of development: selectable phenotypes are not just the adult ones (if any), but all phenotypic conditions throughout development.
3. Moving thus from the role of selection to the origin of selectable phenotypes, AA. are correct in opposing the view that this reduces to random variation at the level of genes or genotypes – but this has been by now accepted by evolutionary biology, although not by all evolutionary biologists individually.
The problem might be if the list of currently acknowledged origins of selectable variation other than random genetic changes must be extended beyond the long lists found in the EES and NCT literature. Maybe it should, but hardly along the lines suggested in this ms. (and in previous literature suggesting similar views and using similar language).
4. The whole ‘argument’ is framed in terms of cells, as if all parts of the argument would uniformly apply to prokaryotes and eukaryotes, to plants and animals, and, more important, to unicells and multicellulars.
5. AA. would offer a model of Cognition Based Evolution, but fail to offer even a single example of what an intelligent cell would measure or learn. Indeed, these cells float in a largely unstructured ‘environment’, where the only faintly visible presences would be other cells.
6. What is gained by calling a cell ‘intelligent’? Without further explanation, what does this add to describing a cell’s behavior in terms of plasticity, or of adaptivness?? Are all machines with homeostatic behavior worth be called intelligent?
There are a number of lesser but not minor problems with this ms. E.g.:
7. A very disputable characterization of Neodarwinism:
Lines 43-46: “Neodarwinism is centered within several enduring pillars: a.) evolution is primarily due to random genetic variations b.) such variations are subject to differential selection across a fitness landscape c.) the resulting process of descent though modification is necessarily gradual and d.) the target of selection is the visible macroorganic form [1,2].”
This referee does not want to defend Neodarwinism without further qualification – but at least wants to stress that the description given in lines 43-46 may apply to some traditional textbook definitions, but does not correspond to current ‘non-revolutionary’ perspectives on evolution still compatible with Darwin’s views.
8. Here and there, there are conspicuous errors in categorization. For example
Lines 84-85: “all multicellular eukaryotes are holobionts comprised by 84 participants of each of the four domains (Prokaryota, Archaea, Eukaryota, Virome)”
Here, AA. list four items as domains (the highest entities in a taxonomy of living forms), but only three (Prokaryota, Archaea, Eukaryota) are in fact domain names, but virome is not. E.g., in Wikipedia:
“Virome refers to the assemblage of viruses that is often investigated and described by metagenomic sequencing of viral nucleic acids that are found associated with a particular ecosystem, organism or holobiont.”
Moreover, since the recognition of Archaea as a distinct domain, the remaining prokaryotes are much better called Bacteria or Eubacteria rather than Prokaryota.
9. Summing up, what is described in this paper could be rephrased in terms of memory (non just in form of genes, of course), variation and selection. But this requires paying attention to history and environment, of which in the whole long ms. there is hardly a presence. Giving intelligence to cells does not seem to provide any appreciable advantage over more sober views of life.

Author Response

Response to Reviewer 3-Cells

The co-authors are grateful for this deeply considered review and even the more so that it is acceded by this  knowledgeable reviewer that “ evolution is not just a history of changes in allelic or genotypic frequencies, or a history of competition to the exclusion of other kinds of interactions, including mutually advantageous ones.”  Please see our responses below to each criticism in brackets with the questions in italics. [Note bene: the lines referenced below pertain to the Word format required for the Cells re-submission.]

  1. defend the sensible view that evolution is not just a history of changes in allelic or genotypic frequencies, or a history of competition to the exclusion of other kinds of interactions, including mutually advantageous ones. In this respect, this is in agreement with a variety of criticism of reductionist and strictly gene-centric Neodarwinism – a criticism expressed in the last decades by many authors under different umbrella terms, such as Extended Evolutionary Synthesis (EES), Niche Construction Theory (NCT) and others, as listed also in this ms., but not discussed.

[The authors agree with the reviewer that there have been many productive amendments to standard Neodarwinism. Indeed, Section 2 does offer a necessarily brief summary of prevalent opinion about the nature of evolutionary biology as it had been and still is generally represented in the literature. Our intent is to honor prior advances and integrate them within a new framework, e.g. niche construction which figures prominently in our manuscript. We hope that the reviewer will agree that given the length of the manuscript that is necessitated in its  vigorous defense of an unfamiliar narrative, that a more exhaustive review of prior thinking is not practical.]

  1. Similar to other views that take more or less explicitly distance from Neodarwinism, AA. seem to deny, or at least strongly downplay, the role of selection. Selection, however, could easily explain stories of adaptation that the AA. would explain otherwise. Why not selection? E.g.:

Lines 256-267: “Collective biofilms represent a predominant form of prokaryotic life since they permit collective engineering as effective form of cellular problem-solving. In Bacillus subtilis biofilms, the participating cells can sub-specialize to elaborate an optimal extracellular matrix for either cell surface adhesion or colony mobility [48]. The specific composition of this extracellular matrix must differ to meet exact environmental and surface conditions to permit either adhesion or motility. In order to produce both, constituent cells partner in specialized activities that completely differ from those that they exhibit within their free living form [49]. These are consensual actions. There are high levels traded resources and some participants accept the voluntary loss of some cellular functions to support the collective whole [50,51]. Each of these united cellular forms with their exclusive and different outputs represents a type of phenotype and each variety has its differential discrete function, whether to stick or to migrate.
In this case, and more generally, the critical issue is not selection, but the origin of selectable variation. Selection, for example, can operate on partnerships (between cells of the same or different kinds) with different fitness in a given environment. Same as for any biological systems considered from the point of view of development: selectable phenotypes are not just the adult ones (if any), but all phenotypic conditions throughout development.

[The co-authors agree that natural selection is an absolute requirement for evolution and believe that they do make that apparent in the manuscript. For example, we state:

“Consequently, selection can now be assessed as an over-arching post-facto filter of antecedent natural cellular engineering solutions to cellular problems. Selection imposes continuous cellular-organismal-environmental complementarity in confrontation with environmental stresses and assures that consensual cellular measurements meet current environmental stresses with sufficient flexibility.  All living forms attest to that success.”

However, we do acknowledge that it is our intent to convince others to reconsider the manner in which selection is widely used in the current literature. As the reviewer might agree, there are many that deem selection's action in terms of quasi-agency. Indeed, the term ' positive selective pressure' that is frequently used implies that selection provides directionality in a 'positive' direction'.  It is one of the specific aims of this manuscript to emphasize that selection can only operate upon preceding variations, to discuss an alternative framework for understanding how such variations might arise, and then reinforce that the imperative of natural selection is its filtering action which enforces continuous complementarity between organisms and the planetary environment. Naturally, not all will agree. That is true even within the reviews of this manuscript. Another reviewer pointed out this aspect of our manuscript as being particularly welcome.]

  1. Moving thus from the role of selection to the origin of selectable phenotypes, AA. are correct in opposing the view that this reduces to random variation at the level of genes or genotypes – but this has been by now accepted by evolutionary biology, although not by all evolutionary biologists individually.
    The problem might be if the list of currently acknowledged origins of selectable variation other than random genetic changes must be extended beyond the long lists found in the EES and NCT literature. Maybe it should, but hardly along the lines suggested in this ms. (and in previous literature suggesting similar views and using similar language).

[With the greatest respect, the co-authors fully understand that there will be others that disagree with the manner in which phenotype arises and it is our intent to enlarge the discussion of the range of possibilities that might account for this level of complexity. It is our belief that our proposal of non-random genomic mechanisms that are deployed in the context of cells that can purposely act in collective manners to enable phenotypes is a useful addition to the ongoing debate about the origination of creative phenotypes. We certainly do not expect that this will end that debate. Indeed, it is our hope that the debate, which remains unsettled, will be encouraged. To that end, as an unfamiliar narrative that is supported with appropriate citations, we hope that it will able to achieve scrutiny by the critical Cells readership.]

  1. The whole ‘argument’ is framed in terms of cells, as if all parts of the argument would uniformly apply to prokaryotes and eukaryotes, to plants and animals, and, more important, to unicells and multicellulars.

[Respectfully, and although our intent is to concentrate of the knotty issues of eukaryotic multicellular evolution, there are aspects of this manuscript which do attend to this concern. For example, Section 3,  'Unicellular insights into biological variation 'does address many aspects of manner in which the unicellular realm varies in continuous response to environmental stresses.]

  1. would offer a model of Cognition Based Evolution, but fail to offer even a single example of what an intelligent cell would measure or learn. Indeed, these cells float in a largely unstructured ‘environment’, where the only faintly visible presences would be other cells.

[ With the greatest respect, the co-authors believe that they offer several examples of natural learning, such as the interactions between donor and recipient cells to effect R plasmid exchange, or the patterns of cellular differentiation and sub-specialization  of Bacillus subtilis that permit differential colonial motility or adhesion. Further yet, the direct contention being made in the manuscript is that the conjoint measurement of information that cells use to sustain mixed-species multicellular tissue ecologies and then link them into the complexity of entire seamlessly functioning holobionts represents a replete example of cognitive assessment and deployment of information as learning.]

  1. What is gained by calling a cell ‘intelligent’? Without further explanation, what does this add to describing a cell’s behavior in terms of plasticity, or of adaptivness?? Are all machines with homeostatic behavior worth be called intelligent?

  2. [We thank the reviewer for making this point, since it is so central to our thought processes and undergirds a substantial part of Cognition-based Evolution. We know that the reviewer would be well-versed in the 'life as a machine' perspective, and we are familiar with the extensive literature that amply supports that point of view. Certainly, even before Schrödinger’s seminal contribution, there had been a ceaseless debate arguing life's essential elements. Our perspective defends one approach towards a productive answer. Our initiating premise, and one previously defended by many (Shapiro, Lyon, Ford, Baluska, etc) is that life requires cognition at every level. We further argue that there is ample scientific evidence that cells are cognitive agents and that this faculty is co-terminus with the origin of life, which has also been well-defended. Thus, our basic premise is to accept this reality and then attempt to delineate a constructive further narrative to indicate how such cellular cognition might yield the biological complexity that exists. We make the further point that intelligence is adaptation, mirroring Hawkings well-known dictum. It defaults, from our point of view, that evolution reduces in such circumstances to the further details of just how intelligent adaptations arise from contingent (cognitive) cellular choices to meet environmental constraints. That pathway is through the collective measurement and deployment of information. This is the basis of Cognition-Based Evolution. Crucially, to our judgment, this represents the crux that separates the living from machines, since it centers the differential  within the nature of biological information. In biology, all information is imprecise, as discussed and defended in the manuscript. This is exactly why cells measure information. Precise information requires no measurement. Machines, such as thermostats with memory circuits measure, but never do so within the context of informational ambiguity. Hence, this is specifically, in our view, how the living frame differs from an abiotic one. We are particularly grateful to the reviewer for the opportunity for us to expand on this difficult point, and readily acknowledge that not all would agree.]

There are a number of lesser but not minor problems with this ms. E.g.:

  1. A very disputable characterization of Neodarwinism:
    Lines 43-46: “Neodarwinism is centered within several enduring pillars: a.) evolution is primarily due to random genetic variations b.) such variations are subject to differential selection across a fitness landscape c.) the resulting process of descent though modification is necessarily gradual and d.) the target of selection is the visible macroorganic form [1,2].”
    This referee does not want to defend Neodarwinism without further qualification – but at least wants to stress that the description given in lines 43-46 may apply to some traditional textbook definitions, but does not correspond to current ‘non-revolutionary’ perspectives on evolution still compatible with Darwin’s views.

[The co-authors agree with this criticism. An amendment has been made in our characterization of Neodarwinism, lines 47-49. We would further note that the first paragraph of Section 2, 'Traditional views on the sources of biological variation' does discuss that the Modern Synthesis has been modified and its traditional values argued back. We agree that it would be desirable to have an option of a fuller discussion, however, this manuscript is already deemed quite lengthy, so we ask the reviewer to permit us to omit further elaborations. We do believe that Cell readers would already be generally versed on this issue, but thank the reviewer for raising it.]

  1. Here and there, there are conspicuous errors in categorization. For example
    Lines 84-85: “all multicellular eukaryotes are holobionts comprised by 84 participants of each of the four domains (Prokaryota, Archaea, Eukaryota, Virome)”
    Here, AA. list four items as domains (the highest entities in a taxonomy of living forms), but only three (Prokaryota, Archaea, Eukaryota) are in fact domain names, but virome is not. E.g., in Wikipedia:
    “Virome refers to the assemblage of viruses that is often investigated and described by metagenomic sequencing of viral nucleic acids that are found associated with a particular ecosystem, organism or holobiont.”
    Moreover, since the recognition of Archaea as a distinct domain, the remaining prokaryotes are much better called Bacteria or Eubacteria rather than Prokaryota.

[The co-authors are in complete agreement. Multiple amendments have been made changing 'four domains' to 'the three cellular domains and the virome' at  lines 32-33 in the abstract, lines 94-95,  lines  168-169, lines 1134-1135, and the final sentence of the conclusion.  With some chagrin, this co-author admits  to having engaged in metaphor rather than exact science. Thank you for pointing out this error.]

  1. Summing up, what is described in this paper could be rephrased in terms of memory (non just in form of genes, of course), variation and selection. But this requires paying attention to history and environment, of which in the whole long ms. there is hardly a presence. Giving intelligence to cells does not seem to provide any appreciable advantage over more sober views of life.

[The co-authors hope that our manuscript will be viewed as a presentation of an unfamiliar, but well-defended alternative evolutionary framework.  Astute readers will decide upon its merits and there will be disagreements. However, the concepts of evolution based on the collective action of intelligent cells is a peer-reviewed frame of reference based on multiple papers in many journals. With the greatest respect, the co-authors do not believe that a memory-based system, of which genes are a prime example, is in and of itself sufficient to explain biological complexity. Within our frame, there are two divergent pathways to be considered. Automata can have memory and non-knowing cells, acting as 'living machines',  could interact in endless rounds of selective culling to create biological complexity. On the other hand, knowing, intelligent cells can act to uphold themselves and then, as this manuscript asserts, do so to deploy resources to sustain themselves against an agitating environment in non-random ways. We believe that this latter stance, which requires contingent action at the cellular level, offers a robust explanation for the  emergence of biological complexity. In this frame, cells use memory beyond thermostatic set-points as sources of creativity. It is specifically upon this crux that we have chosen to emphasize the crucial issue of cellular intelligence and are hopeful that the reviewer will permit us to make our case to the discerning Cells readership.]

Round 2

Reviewer 1 Report

The authors have satisfactorily responded to all my questions and resolved all the issues I raised in my previous report.

Author Response

Thank you. The co-authors greatly appreciate your help in improving the manuscript.

Reviewer 3 Report

I appreciate the Authors' extensive and articulated replies to my former criticisms, but some important points have not been adequately addressed,. 

Of the criticisms raised in the first round, those that in my opinion should be further considered by the Authors are listed and briefly discussed.

First round comment 2. – [..] Selection, however, could easily explain stories of adaptation that the AA. would explain otherwise. Why not selection?
Authors’ reply. – [..] the term ' positive selective pressure' that is frequently used implies that selection provides directionality in a 'positive' direction'.
Reviewer’s new comment. – Not at all. No directionality, in teleological sense, is implied by the ncurrent usage of “positive selection pressure.”. Cf. the following definition [from Derbyshire MC (2020) Bioinformatic Detection of Positive Selection Pressure in Plant Pathogens: The Neutral Theory of Molecular Sequence Evolution in Action. Front. Microbiol. 11:644.doi: 10.3389/fmicb.2020.00644]:
Alleles that cause advantageous phenotypes with a greater reproductive rate are said to be under positive selection. Those that cause disadvantageous phenotypes with poorer reproductive rates are said to be under negative selection. Other alleles that have no impact on phenotype (or impacts that do not affect reproductive rate) are said to be neutral (Molles and Sher, 2018).[ citing in turn:: Molles, M. C., and Sher, A. (2018). Ecology: Concepts and Applications. New York,NY: McGraw-Hill Education.]
First round comment 4. - The whole ‘argument’ is framed in terms of cells, as if all parts of the argument would uniformly apply to prokaryotes and eukaryotes, to plants and animals, and, more important, to unicells and multicellulars.
Authors’ reply. - Respectfully, and although our intent is to concentrate of the knotty issues of eukaryotic multicellular evolution, there are aspects of this manuscript which do attend to this concern. For example, Section 3, 'Unicellular insights into biological variation 'does address many aspects of manner in which the unicellular realm varies in continuous response to environmental stresses.]
Reviewer’s new comment. - Quite probably, I was not clear enough in my comment. The point is, that something like “the cell”, intelligent or not, does not exist in nature. Instead, the world is full of cells, in their amazing diversity, in their diverse degree of association with similar cells and/or with cells of different kinds (different genotype and/or phenotype). It is true that all cell share a number of basic properties, but it is also true, and also far from marginal, that they differ in many other respects. What is true of a prokaryote is not necessarily true of a eukaryote. To give an historical example, think of Monod’s operon. Discovered in prokaryotes, should it be also expected in eukaryotes? Etcetera. Or, to consider multicellulars, multicellularity by aggregation, as in cellular slime molds, is not the same as the ordinary multicellularity produced by cells remaining together following mitosis. Moreover, in evolutionary biology we have long abandoned the old habit to imagine that evolution necessarily runs from the simplest to the most complex (whatever ‘complex’ might mean), or, more broadly, according to formal progressive or regressive series, as in the once popular series of the members of the horse family (Equidae). Thus, picking examples freely from cells of very different nature (and phylogenetic origin) does not produce a sound history of evolution, other that by serendipity.
First round comment 5. – [Authors] fail to offer even a single example of what an intelligent cell would measure or learn. Indeed, these cells float in a largely unstructured ‘environment’, where the only faintly visible presences would be other cells.
Author’s reply. - With the greatest respect, the co-authors believe that they offer several examples of natural learning, such as the interactions between donor and recipient cells to effect R plasmid exchange, or the patterns of cellular differentiation and sub-specialization of Bacillus subtilis that permit differential colonial motility or adhesion.
Reviewer’s new comment. - It is difficult to construe these verbal descriptions as example of what an intelligent cell would measure or learn. “Measure” implies a quantity that is measured, like the concentration of a molecule in a solution, a temperature gradient, a codified information; and a cell’s subsystem, or process, by which the measure could be obtained, stored, processed.
Moreover, AA.’s reply does not provide any clear suggestion of what a cell’s environment might be, beyond other cells, of the same or of other “species”, with which it interacts. It is difficult to discuss evolution, even in theoretical terms, without reference of an environment, whose properties can be modelled, or described (measured) by the observer.
First round comment 6. - What is gained by calling a cell ‘intelligent’? Without further explanation, what does this add to describing a cell’s behavior in terms of plasticity, or of adaptivness?? Are all machines with homeostatic behavior worth be called intelligent?
Author’s reply. – [..] We make the further point that intelligence is adaptation [..]
Reviewer’s new comment. - If you accept that “intelligence is adaptation”, you reduce intelligence to a synonym of adaptation. What is gained with this change of term?
First round comment 9 [recorded as 10 in association with the AA.’s reply]. - Summing up, what is described in this paper could be rephrased in terms of memory (non just in form of genes, of course), variation and selection. But this requires paying attention to history and environment, of which in the whole long ms. there is hardly a presence. Giving intelligence to cells does not seem to provide any appreciable advantage over more sober views of life.
Authors’ reply. - Automata can have memory and non-knowing cells, acting as 'living machines', could interact in endless rounds of selective culling to create biological complexity.

Reviewer’s new comment. - Where this true, no cell would probably be around, unless created with its necessary intelligence.
The model of evolution rightly opposed by the AA. is the naïve (unfortunately, still popular, but long abandoned by mainstream evolutionary biology) that reduced evolution to mere play of chance: chance in generating variation, chance in the selective events whose temporal series is a Markovian chain. But biological evolution is a historical events, where the past leaves a trace (memory) that sets conditions for future, even distant, change. Selection ensures that many maladaptive phenotypes are lost along the way.

Round 3

Reviewer 3 Report

I appreciate the strong efforts of the Authors to take this reviewer's remarks in serious consideration and to discuss in detail how/where a line of mutual agreement coulb be obtained, without requiring from either part anything but willingness to keep the dialogue alive.

At this stage, as the ms. has not been modified, my scores for the five questions in the section above are identical to those submitted with my previous report. However, following the further exchange with the Authors, I accept that the paper can be accepted by the Editor for publication, to offer it to the public discussion.